# WEBBRAIN: LEARNING TO GENERATE FACTUALLY CORRECT ARTICLES FOR QUERIES BY GROUNDING ON LARGE WEB CORPUS

## ABSTRACT

In this paper, we introduce a new NLP task – generating short factual articles for queries by mining supporting evidence from the Web. In this task, called WEB-BRAIN, the ultimate goal is to generate a fluent, informative, and factually-correct short article (*e.g.*, a Wikipedia article) for a factual query unseen in Wikipedia. To enable experiments on WEBBRAIN, we construct a large-scale dataset WebBrain-Raw by extracting English Wikipedia articles and their crawlable Wikipedia references. WebBrain-Raw is ten times larger than the previous biggest peer dataset, which can greatly benefit the research community. Besides, we empirically analyze the performances of the current state-of-the-art NLP techniques on WEB-BRAIN and introduce a new framework ReGen, which enhances the generation factualness by improved evidence retrieval and task-specific pre-training for generation. Experiment results show that ReGen outperforms all baselines in both automatic and human evaluations.

## 1 INTRODUCTION

Information acquisition is one of the fundamental daily needs of human beings. Acquiring information from the Web is undoubtedly a convenient and efficient way. However, with the exponential growth of the Web, information on the Web becomes scattered and evolves quickly, making it challenging for users to acquire the expected information quickly. As a result, Wikipedia articles become the best bet for most users when searching answers for factual queries on the Web (Singer et al., 2017). The reason is that Wikipedia articles provide credible content in which most claims can be supported by references from reputable sources. While Wikipedia is a good source of answers for factual queries, the need for manual editing (crowd-sourcing and editor checking) curbs its growth of coverage on a broader range of information needs. What if Wikipedia articles could be automatically generated?

In this paper, we introduce a new task, WEBBRAIN, exploring the capacity of generating short factual articles for queries via a large web corpus. **Given a factual query, the goal of the task is to enable a system to mine supporting evidence from the Web and generate a short factual article in which the claims are supported by the mined evidence** (defined in Section 3.1). One of the potential generation targets for WEBBRAIN is the first section of a new Wiki page, based on which we can further explore generating long factual articles (*e.g.*, a complete Wiki page). WEBBRAIN can be greatly helpful in various scenarios, including generating Wiki pages for new entities, intelligent writing assistance, knowledge-intensive QA, etc. WEBBRAIN's goal is considered one of the ultimate goals of the future search engine (Metzler et al., 2021). Figure 1 illustrates a case of our WEBBRAIN.[1]

To establish the data foundation of WEBBRAIN, we construct a large-scale dataset, WebBrain-Raw, from scratch by extracting all English Wikipedia articles and all the corresponding reference articles. To the best of our knowledge, WEBBRAIN-Raw is the biggest dataset sourced from Wikipedia (about $10\times$ larger than the previous biggest peer WikiSum (Liu et al., 2018), introduced in Section 3.2). Along with WEBBRAIN, we empirically investigate the ability of the current state-of-the-art techniques and conclude that most current models lack the ability to correctly cite references and

---

[1] the text generation result is obtained via OpanAI's GPT3 API: `https://beta.openai.com/`

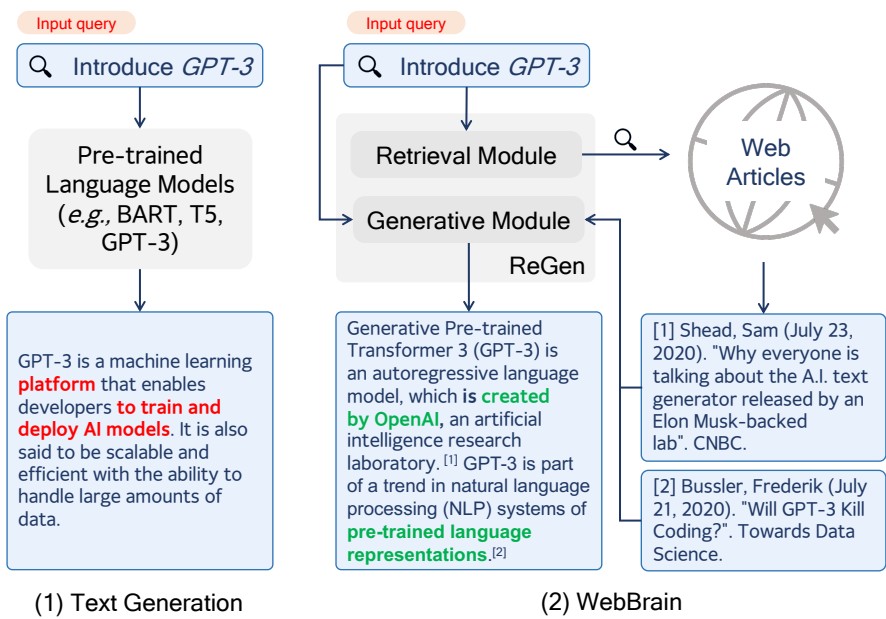

Figure 1: Comparison between text generation task and WEBBRAIN.

generate factual descriptions. Based on the best current models (SPLADE+FiD+BART) (Izacard & Grave, 2021; Formal et al., 2021a), we propose a new framework, ReGen, which enhances the factual correctness of generation by (1) controlling the topic consistency of the retrieved supporting evidence; (2) introducing the citation mark into the generation process to indicate the source of a referenced claim; (3) warming-up the generation model by pre-training with fact-oriented self-supervised tasks (described in Section 4).

Current tasks that are similar to WEBBRAIN include: (1) text generation with generative pre-trained language models (*e.g.*, GPT-3) (Brown et al., 2020); (2) retrieval-augmented QA (*e.g.*, RAG (Lewis et al., 2020b), GopherCite (Menick et al., 2022), REALM (Guu et al., 2020)); (3) multi-document summarization (MDS) (Liu et al., 2018; Liu & Lapata, 2019); (4) WebGPT (Nakano et al., 2021).

WEBBRAIN distinguishes itself from the existing tasks in the following aspects: (1) Existing generative models generate text solely depending on the implicit knowledge stored in the model's parameters, which are prone to generating factually incorrect statements, a phenomenon commonly called *hallucination*. In contrast, our WEBBRAIN aims to generate factual statements depending on the supporting evidence mined from the Web; (2) Retrieval-augmented QA utilizes the retriever to enhance answering specific questions whose answers are usually a text span or a single sentence. Given a factual query, our WEBBRAIN explores capturing all the knowledge available on the Web to generate a comprehensive and accurate short article; (3) WEBBRAIN has a more complex pipeline than the MDS task, which focuses on summarizing documents that are already prepared. WEB-BRAIN requires models to mine useful evidence before generating the factual article; (4) WebGPT mimics the behavior of a human by browsing the Web to answer a specific question. This involves many challenges that are difficult to solve (*e.g.*, collecting training data by recording human behaviors). Instead, WEBBRAIN grounds itself on a large number of Wikipedia articles that are already created and edited by crowds, which makes the task much more realistic.

In summary, our contributions are threefold: (1) we introduce a new task, WEBBRAIN, dedicated to answering factual queries by generating short factual articles based on evidence retrieved from a large web corpus; (2) we construct a large-scale dataset, WebBrain-Raw, to evaluate the potential of WEBBRAIN. WebBrain-Raw is ten times larger than the previous biggest peer dataset; and (3) we empirically analyze the performance of current state-of-the-art techniques on WEBBRAIN, and propose our factuality-enhanced framework, ReGen, which outperforms all baselines through both automatic and human evaluation.

## 2 RELATED WORK

**Pre-trained Language Models** Pre-trained language models (PLMs) have been widely applied to various natural language processing (NLP) tasks and have achieved outstanding performance. In general, these models are first *pre-trained* on a large-scale corpus, and then *fine-tuned* on downstream datasets for specific tasks. By pre-training, the model can learn effective language representations, thus improving its performance on downstream tasks. Typical paradigms of PLMs include masked LM (Devlin et al., 2019; Zhang et al., 2019; Sun et al., 2019), left-to-right LM (Radford et al., 2019; Black et al., 2021), prefix LM (Dong et al., 2019; Bao et al., 2020), and encoder-decoder LM (Song et al., 2019; Raffel et al., 2020; Lewis et al., 2020a). Masked LMs are generally most suitable for natural language understanding or analysis tasks, *e.g.*, text classification and natural language inference. The other three types of PLMs can be naturally used for text generation.

**Wikipedia-related Tasks** Wikipedia is a popular knowledge source for NLP tasks that aim to utilize external knowledge. Examples include knowledge-grounded conversation (Dinan et al., 2019), fact checking (Thorne et al., 2018), open-domain QA Kwiatkowski et al. (2019), and slot filling El-Sahar et al. (2018) etc. Petroni et al. (2021) propose a benchmark for those knowledge-intensive language tasks (KILT) in which all tasks apply a snapshot of Wikipedia as the knowledge base. Compared to the KILT, other Wikipedia-related tasks focus more on Wikipedia itself. Liu et al. (2018) construct a dataset WikiSum from Wikipedia to enable multi-document summarization (MDS). The MDS task is then further explored by Liu & Lapata (2019) with a hierarchical model and Perez-Beltrachini et al. (2019) with a topic-guided structure. WikiWrite (Banerjee & Mitra, 2016) proposed a systematic solution to constructing Wiki pages by assigning retrieved content to different topical sections. Fruit (Iv et al., 2022) explores updated information on existing Wiki pages. Recently, Piktus et al. (2021) construct a dataset Sphere from CCNet (Wenzek et al., 2020), using which they explore refining the citation quality of Wikipedia pages.

**Retrieval-Augmented Text Generation** Pre-trained language models are prone to generating hallucinations (*i.e.*, factually incorrect statements) (Vinyals & Le, 2015; Koehn & Knowles, 2017; Rohrbach et al., 2018; Raunak et al., 2021). Retrieval-augmented text generation, as a new text generation paradigm, can be a possible way to alleviate this problem. Compared with its generation-based counterpart, this new paradigm can reduce the reliance on storing enormous knowledge in parameters by providing more references. Retrieval-augmented text generation has been widely applied to many NLP tasks, such as dialogue generation (Weston et al., 2018; Zhu et al., 2020), machine translation (Gu et al., 2018; Cai et al., 2021), and open-domain question answering (Lewis et al., 2020b; Guu et al., 2020). In this work, we formalize a new retrieval-augmented text generation task – WEBBRAIN with two new features: (1) The retrieval base is a large-scale open-domain web corpus; and (2) The target is to generate a natural, informative, and factual text rather than a short span. Both features make the task extremely challenging. Our experiments will show that, though existing models can be applied to this task, there is still a large space for research and exploration.

## 3 TASK: WEBBRAIN

### 3.1 TASK DEFINITION

Formally, given an input query $q$, our task is to generate a short article $T_q$ for $q$ grounded by prior knowledge $\Theta$. We have:

$$T_q^* = \arg\max p(T_q|q, \Theta). \tag{1}$$

For a pre-trained language model solely, the knowledge prior $\Theta$ is stored in the PLM's internal parameters. For WEBBRAIN, the knowledge prior $\Theta$ contains the implicit language knowledge $\theta$ stored in the PLM's parameters and the explicit knowledge $\gamma$ defined by the supporting evidence $V$, which is mined from an external large web corpus $C$. In practice, $C$ can be an off-line processed large web corpus, an online search engine API, or a combination of both. Intuitively, coupling $\theta$ and $\gamma$, $\Theta = (\theta, \gamma)$ can be characterized by the strong language capacity from the PLM and the factual constraints from the supporting evidence.

Supposing the generated $T_q$ contains $n$ sentences $(t_1, \cdots, t_n)$ and the retrieved supporting evidences $V = (v_1, \cdots, v_k) \in C$, we expect each sentence $t$ would be supported by a $V$'s subset $V^\tau$, denoted

Table 1: Datasets built on Wikipedia. WEBBRAIN-Raw is significantly larger than existing datasets.

| Dataset | # Wiki Pages | # Refs | Status | Storage Size |
|---|---|---|---|---|
| WikiSum (Liu et al., 2018) | 2.3M | 87M | Need crawling | 300GB |
| WikiCatSum (Perez-Beltrachini et al., 2019) | 0.17M | 23.5M | Ready | 4.8GB |
| Hiersumm (Liu & Lapata, 2019) | 1.66M | - | Ready | 6.9GB |
| WebBrain-Raw | 14.86M | 259.5M | Ready | 2.8TB |

as $t^{\leftarrow V^\tau}$. More precisely, we have:

$$T_q^* = (t_1^{\leftarrow V^{\tau_1}}, \cdots, t_n^{\leftarrow V^{\tau_n}}) = \arg\max p(T_q | q, \theta, \gamma), \quad \gamma := V \in C. \tag{2}$$

Note that when the generated sentence only relies on the PLM's language knowledge $\theta$ (*e.g.*, common sense generation), the corresponding supporting evidence is $V^\tau = \emptyset$.

## 3.2 DATA COLLECTION

Wikipedia is one of the most inclusive and widely-used encyclopedias on the Web. By the nature of an encyclopedia, Wikipedia can be regarded as a large collection of factual articles on various topics (*e.g.*, entry of a Wikipedia page). The factual content in Wikipedia is generated by crowd-sourcing, and most claims can be supported by the citations in the *References* section. Due to such nature, Wikipedia is the most appropriate resource to leverage for WEBBRAIN. The specific reasons are: (1) Wikipedia's format conforms to the WEBBRAIN task's definition in Eq. (2). We can view the Wiki page title as a factual query $q$, the citations (or parts of them) as the retrieved supporting evidence $\gamma$, and the Wiki article as the target factual article $T_q^*$; (2) Wikipedia has a huge volume of references which are usually reputable sources. Using the references, we can build a large web corpus of good quality as the retrieval source; (3) Wikipedia covers a wide range of topics of common interest, which provides a good knowledge basis for models to generalize to open-domain queries.

Many previous studies collected Wikipedia data for different tasks (Liu et al., 2018; Liu & Lapata, 2019; Perez-Beltrachini et al., 2019). However, none of these datasets provides full-size Wikipedia pages and the corresponding references. To lay a good data foundation for WEBBRAIN, we build a large-scale dataset from scratch by extracting all Wikipedia pages and all crawlable references. Table 1 shows the statistics of different datasets. In the Supplementary Material, we attach 500 raw data samples and the corresponding experiment data for demonstration. The full datasets we construct for WEBBRAIN will be released upon acceptance of this paper.

**Data Cleaning** As shown in Table 1, the WebBrain-Raw data contains full-size Wiki pages and a huge volume of references, which are open-domain web pages. Regarding data quality, the Wiki pages are good due to their server stability and format consistency. However, the references contain much noise or are useless for our task. For example, they may be images, tables, or empty content, etc. To guarantee data quality, we process the WebBrain-RAW dataset in the following steps:

Table 2: Statistics of data for experiments.

| | WebBrain-R | WebBrain-G |
|---|---|---|
| # Queries | 2.74M | 12.32M |
| # Ref. passages | 3.20M | 12.61M |
| # Tokens / Query | 3.2 | 2.9 |
| # Tokens / Passage | 237.5 | 250.0 |
| # Tokens / Target | - | 108.6 |
| # Training | 4.46M | 12.30M |
| # Validation | 0.2M | 0.5M |
| # Test | 88,935 | 24,546 |

(1) **Wiki Article Cleaning**: for Wiki pages, we remove the Wiki template[2], special symbols (*e.g.*, ASCII symbols), and other invalid text[3].

(2) **Reference Cleaning**: for reference, we filter out articles that are shorter than 16 words or non-English token ratio $> 0.3$. We remove invalid text such as those containing only HTML tag, unusually long sentences (*e.g.*, $> 256$ tokens), text tables and URLs, etc. Furthermore, as the citations

---

[2]See details in `https://en.wikipedia.org/wiki/Wikipedia:WikiProject_Templates`
[3]We observe some template text that may not be useful for model training (*e.g.*, `You can help Wikipedia by expanding it.`)

may be missing (uncrawlable or filtered out) or incorrectly cited (Piktus et al., 2021), we define the following metric to calculate the term recall of a reference to choose "reasonable" references:

$$P_{\text{ST}} = \frac{|(\mathcal{Q} \cup \mathcal{T}) \cap \mathcal{V} - \mathcal{S}|}{|\mathcal{T} - \mathcal{S}|}, \tag{3}$$

where $\mathcal{Q}$ is the term set of a query $q$, $\mathcal{T}$ is the term set of a Wiki article sentence $t^{\leftarrow V^{\tau}}$ that cites a reference set $V^{\tau}$, $\mathcal{V}$ is the term set of $V^{\tau}$, and $\mathcal{S}$ is a set of stop words. Intuitively, if more terms in the sentence claim $t$ can be found in the references $V^{\tau}$, the references would be more useful. Therefore, we only keep references that have $P_{\text{ST}} > \rho$.

**Reference Passage Selection** The original Wiki articles and reference articles tend to be very long. To adapt the capacity of most pre-trained language models (*e.g.*, BERT has a 512-token limit), we use the first section of Wiki articles as the generation target, and we select the passage from the reference article with the highest $P_{\text{ST}}$ value as the supporting passage. Specifically, we first split a reference article by sentences and concatenate the sentences into passages (stride $= 1$, max_length $= 256$), then compute the $P_{\text{ST}}$ value for each passage and keep the highest one.

**Dataset Generation** As defined in Eq. (2), given a query $q$, the WEBBRAIN task aims to retrieve the supporting evidence $V$ from a web corpus $C$, to generate a factual article $T_q$. To enable training the in-domain retriever and generator for WEBBRAIN, we generate two separate datasets, WebBrain-R(etriever) and WebBrain-G(enerator), from WebBrain-Raw. For WebBrain-G, we keep five supporting passages for each query [4], and retrospectively assign reference marks to all unmarked sentences in the Wiki article by measuring their $P_{\text{ST}}$ value. We append a " [0] " mark to the sentence end for these fail to match any references. For WebBrain-R, we use the supporting passages as the positive and randomly select four negative passages from the top 30-50 retrieved results via a BM25 engine. Table 2 shows the statistics of the two datasets.

# 4 FRAMEWORK: REGEN

## 4.1 RETRIEVER

By the definition of WEBBRAIN in Eq. (2), ReGen starts from a retriever that mines the supporting evidence $V$ from the large web corpus $C$. ReGen utilizes a parametric sparse lexical model SPLADE (Formal et al., 2021a;b) as the retriever, which encodes a sequence $s = (w_1, \cdots, w_n)$ into a sparse lexical representation $\mathbf{s} \in \mathbb{R}^{|D|}$ by predicting token-level importance in WordPiece vocabulary size (*e.g.*, $|D| = 30522$). Specifically, $\mathbf{s}$ is computed based on the dense representation $(\mathbf{h}_1, \cdots, \mathbf{h}_n)$ output by the underlying PLMs:

$$\mathbf{w}_i = \mathbf{EWh}_i^{\top} + \mathbf{b}, \quad i \in [1, n], \quad \mathbf{s} = \max_{i \in [1,n]} \log(1 + \text{ReLU}(\mathbf{w}_i)), \tag{4}$$

where $\mathbf{W} \in \mathbb{R}^{768 \times 768}$ and $\mathbf{b} \in \mathbb{R}^{|D|}$ are a trainable transition matrix and a bias, respectively, and $\mathbf{E} \in \mathbb{R}^{768 \times |D|}$ refers to the PLM's input embedding matrix. In the training phase, given a query $q$ and a document $d$, the ranking score $g(q, d)$ is computed by the dot product of the their representations $\mathbf{s}_q$ and $\mathbf{s}_d$.

Previous studies that utilize a retriever in the generation loop usually apply the traditional term-match-based retriever (*e.g.*, BM25), dense passage retriever (*e.g.*, DPR), or a hybrid retriever (Karpukhin et al., 2020; Lewis et al., 2020b; Izacard & Grave, 2021; Glass et al., 2022) We build ReGen's retriever based on SPLADE because of the following reasons: (1) Factual queries in WEBBRAIN are usually open-domain and entity-centric, while recent work observed that DPR tend to perform poorly for entity-centric queries or out-of-domain queries (Sciavolino et al., 2021; Xiong et al., 2021; Thakur et al., 2021). SPLADE, on the contrary, inherits the good properties of sparse retrievers such as exact-entity match and generalizability, making it suitable for WEBBRAIN. (2) SPLADE also inherits the contextualized semantics of the underlying PLMs, enabling the query encoder to perform contextualized query expansion (Formal et al., 2021a), which alleviates the vocabulary mismatch problem of BM25. (3) Empirical results also prove the effectiveness of SPLADE in WEBBRAIN (see Section 5.3).

---

[4]statistics shows that the first section of 96%+ Wiki pages has equal or less than five references.

**Ranking Loss**   Generally, given a query $q$, a positive document $d^+$ and a set of negative documents $d^-$, the optimization goal of a retriever is to maximize the probability:

$$\mathbf{P}(q, d^+, \mathcal{D}) = \frac{\mathbf{e}^{f(q,d^+)}}{\mathbf{e}^{f(q,d^+)} + \sum_{d^- \in \mathcal{D}} \mathbf{e}^{f(q,d^-)}}, \tag{5}$$

where $\mathcal{D}$ represents the entire corpus. We use the mined negative passages and the in-batch negative samples as $d^-$. We train the model with a contrastive LCE loss (Gao et al., 2021).

**Asymmetric Sparsity**   We introduce two sparsity constraints to control SPLADE's sparsity: L1 regularization and FLOPS regularization (Paria et al., 2020). For the query encoder and document encoder, we apply asymmetric sparsity regularization. We use L1 regularization with a small weight $\lambda_q$ for the query encoder and FLOPS regularization with a big weight $\lambda_d$ for the document encoder. Thus, the final loss for SPLADE becomes:

$$\mathcal{L} = \mathcal{L}_{\texttt{rank}} + \lambda_q \mathcal{L}_{\texttt{L1}} + \lambda_d \mathcal{L}_{\texttt{FLOPS}}. \tag{6}$$

We apply the asymmetric sparsity because: (1) a small $\lambda_q$ lowers the sparsity of the query encoder, enabling the query encoder to generate semantically-rich query representation; (2) the FLOPS regularization is not sensitive to the number of documents in a batch so we can compute $\mathcal{L}_{\texttt{FLOPS}}$ on both positive and negative samples; (3) a big $\lambda_d$ increases the sparsity of the document encoder, making the off-line document index more efficiently, and producing a self-denoising effect – more unimportant tokens tend to be ignored.

**Topic Consistency and Diversity**   For a document pair $(d_m, d_n)$ and their $|D|$-size sparse representations $(\mathbf{s}_m, \mathbf{s}_n)$, we measure their topic distance by:

$$d_T(d_m, d_n) = \texttt{distance}(\mathbf{s}_m, \mathbf{s}_n) = \frac{|\mathbf{s}'_m - \mathbf{s}'_n|}{|\mathbf{s}'_m| + |\mathbf{s}'_n|}, \quad \mathbf{s}'[s_j > \mu] = 1, \quad j \in |D|, \tag{7}$$

where $\mathbf{s} = (s_1, \cdots, s_j)$, $\mathbf{s}'$ is a binary variant of $\mathbf{s}$, and $\mu$ is a threshold to eliminate tokens with low importance. For $(d_m, d_n)$, when $d_T \to 0$, the two documents tend to have the same important tokens and are likely to be redundant; whereas when $d_T \to 1$, the two documents tend to have non-overlapping important tokens and are likely topically irrelevant. In the ReGen framework, we propose a strategy to maintain topic consistency and avoid redundancy: Considering $k$ sorted retrieved evidence documents $(d_1, \cdots, d_k)$, we first filter out the documents $d_i$ that have $d_T(d_1, d_i) > 0.9, i \in [2, k]$ (topically-irrelevant to the top-1 document). We then remove documents $d_i$ that have $d_T(d_i, d_j) < 0.1, j \in [1, i-1]$ (redundant to fore-rank documents).

## 4.2   GENERATOR

ReGen employs a sequence-to-sequence model with a Fusion-in-Decoder (FiD) structure (Izacard & Grave, 2021) as the generator. The generator takes the query and references as input and generates a factual description. More precisely, given a query $q$ and a set of $k$ references $\{v_1, v_2, \cdots, v_k\}$ returned by the retriever, we add special tokens before the query/reference and concatenate the query with each reference as:

$$x_i = \texttt{[query]}\, q\, \texttt{[ref\_i]}\, v_i. \tag{8}$$

These sequences are processed independently by the encoder as:

$$\mathbf{h}_i = \texttt{Encoder}(x_i), \quad i = 1, 2, \cdots, k. \tag{9}$$

Finally, the decoder conducts attention over the concatenation of the representations of all the sequences, and outputs a text sequence $T^*$ with auto-regressive mechanism:

$$T^* \sim \texttt{Decoder}(T^*, \mathbf{h}) = \prod_{n=1}^{|T^*|} p(T^*|T^*_{<n}, \mathbf{h}), \qquad \mathbf{h} = [\mathbf{h}_1; \mathbf{h}_2; \cdots; \mathbf{h}_k]. \tag{10}$$

The model can perform evidence fusion in the decoder and generate more factual claims with the references. The generator is optimized by minimizing the negative log-likelihood function of the ground-truth text:

$$\mathcal{L}_{\text{NLL}}(x, T^*) = -\sum_{n=1}^{|T^*|} \log p(T^*_n|T^*_{<n}, x). \tag{11}$$

**Model Warm-Up for Sentence-Reference Matching**    In our task, the model is trained to generate claims for a query based on several references. This task is difficult as the model should first find clues from various references and then determine which part is the most relevant in the current generation status. To facilitate the model's capability of extracting key information from the reference, we design a model warm-up stage. Specifically, for a target text, we break it into multiple sentences and extract the ones that are marked with references. As a result, we can obtain several ⟨reference, sentence⟩ pairs. Then, we concatenate the query with the reference, and train the model to generate the corresponding sentence. This warm-up stage has **two advantages**: (1) The task is to generate a sentence with a single reference. Compared with the final generation task (*i.e.*, generating multiple sentences with multiple references), this warm-up task is much simpler. Therefore, the overall training follows a curriculum learning paradigm, *i.e.*, learning from easy tasks to hard tasks. This learning paradigm has been shown to be effective in improving performance (Bengio et al., 2009; Hacohen & Weinshall, 2019). (2) Existing models often apply general tasks for pre-training. For example, BART (Lewis et al., 2020a) uses several sequence denoising tasks for pre-training. These tasks have different objectives from text generation. Our proposed warm-up task is also a text generation task, so it can help the model mitigate the gap of tasks between pre-training and fine-tuning.

## 5 EXPERIMENTS

### 5.1 SETTINGS

ReGen applies SPLADE as the basis of retriever Formal et al. (2021a), improves the vanilla SPLADE with a training strategy of asymmetric sparsity, and uses a ranking strategy considering documents' topic consistency and diversity. For comparison, we report the performance of the vanilla SPLADE, DPR (Karpukhin et al., 2020) and BM25. We use RetroMAE as the underlying PLM of DPR, which is shown to be a state-of-the-art model in many retrieval benchmarks (Liu & Shao, 2022). For the generator, we use FiD+BART as the backbone model. As a comparison, we report the performance of BART (Lewis et al., 2020a), GPT-2 (Radford et al., 2019), and vanilla FiD+BART (Izacard & Grave, 2021). All models are initialized with the off-the-shelf checkpoints provided by HuggingFace (Wolf et al., 2019). The implementation details are in Appendix A.

To evaluate the performance, we employ several automatic metrics as follows and perform a human annotation: **BLEU** (Papineni et al., 2002), **METEOR** (Banerjee & Lavie, 2005), **ROUGE** (Lin, 2004), and **CIDEr** (Vedantam et al., 2015): These are metrics based on $n$-gram overlapping between generated text and ground-truth text. A higher value indicates the generated text is more similar to the ground truth. We compute the metrics using the `nlg-eval` package (Sharma et al., 2017). **QAGS** (Wang et al., 2020) and **Triple Score** (Goodrich et al., 2019) are two metrics for evaluating factualness. We also compute them between generated text and ground-truth text.[5] For QAGS, several questions are generated from the generated text, which are then answered by both the ground-truth text and the generated text. A higher value reflects that more identical answers are obtained, which further suggests that the generated text is more factual. To compute Triple Score, the relations in both generated text and ground-truth text are first extracted by OpenIE (Angeli et al., 2015). Then, the score is computed by the precision of relations in the generated text. A higher score indicates that more relations are correctly generated. We compute QAGS and triple score by `FactSumm` package (Heo, 2021). As for human annotation, we compare the result of our ReGen with that of baselines concerning fluency, informativeness, and faithfulness. A detailed description of the annotation criteria is given in Appendix B.

### 5.2 OVERALL RESULTS

The results of various models are shown in Table 3 and Table 5. As can be seen, our ReGen model (in both sizes) achieves the best results on all metrics. This clearly demonstrates the effectiveness of our framework. Specifically, (1) Compared with BART and GPT-2, FiD and ReGen achieve better performance, especially on factualness metrics (QAGS and triple score). This indicates that the knowledge from the Web corpus is critical for WEBBRAIN task. (2) Large models can always perform better than base models, but the main improvement is made on language quality (*e.g.*,

---

[5]These scores can also be computed between generated text and reference, but how to combine the scores based on multiple references is still under-explored.

Table 3: Performance of different models on WebBrain-G. The best results are in bold.

| Model | BLEU-1 | BLEU-4 | METEOR | ROUGE-L | CIDEr | QAGS | TripleScore |
|---|---|---|---|---|---|---|---|
| BART (Base) | 8.16 | 2.98 | 9.28 | 23.08 | 38.30 | 28.49 | 10.91 |
| BART (Large) | 11.63 | 4.55 | 10.49 | 24.45 | 45.54 | 28.61 | 12.47 |
| GPT-2 | 9.35 | 2.85 | 8.57 | 20.22 | 25.09 | 25.61 | 7.30 |
| FiD+BART (Base) | 11.20 | 4.72 | 12.60 | 27.42 | 51.32 | 39.88 | 15.89 |
| FiD+BART (Large) | 15.90 | 6.89 | 14.03 | 28.64 | 57.06 | 39.52 | 17.36 |
| ReGen (Base) | 12.11 | 5.26 | 12.98 | 27.81 | 52.92 | 40.00 | 16.19 |
| ReGen (Large) | **17.97** | **8.82** | **15.31** | **30.09** | **62.39** | **40.32** | **19.43** |

Table 4: Retrieval (left) and generation (right) performance of ReGen (Large) with different retrievers. Passages for generation are retrieved from the full passage set. The best results are in bold.

| Model | R@1 | R@5 | MAP | ‖ | BLEU-1 | METEOR | ROUGE-L | CIDEr | TripleScore |
|---|---|---|---|---|---|---|---|---|---|
| ReGen | **0.399** | **0.746** | **0.550** | ‖ | **27.81** | 14.89 | **22.00** | **19.40** | **13.68** |
| + BM25 | 0.290 | 0.558 | 0.410 | ‖ | 27.16 | 14.54 | 21.38 | 17.32 | 12.98 |
| + DPR | 0.228 | 0.390 | 0.301 | ‖ | 24.57 | 13.17 | 19.82 | 15.21 | 10.74 |
| + SPLADE | 0.389 | 0.724 | 0.536 | ‖ | 25.60 | **14.98** | 20.71 | 14.24 | 12.35 |
| Oracle | - | - | - | ‖ | 17.97 | 15.31 | 30.09 | 62.39 | 19.43 |

BLEU and ROUGE) rather than factualness (*e.g.*, QAGS). This is consistent with our assumption that external knowledge is hard to be "remembered" in model parameters. (3) Compared with FiD, we design a warm-up stage for ReGen model. This strategy is effective regardless of model size but works better for a larger model. It suggests that larger models with more capacity are promising for better results, but their potential needs to be activated through well-tailored tasks.

## 5.3 DISCUSSION

**Impact of Retriever**   To explore how different retrievers impact the end-to-end performance of ReGen, we replace ReGen's underlying retriever with BM25, DPR (RetroMAE), and SPLADE. To simulate real-world scenarios, for generation, all passages are retrieved from a full passage set (without additional selection). From the results shown in Table 4, we can make the following observations: (1) Compared to using oracle references, the BLEU-1 score increases while the other metrics decrease, indicating that the retrieved knowledge enriches the informativeness of the generation but undermines its quality. This implies that the retrieval quality is crucial for the factuality of generation, and how to retrieve helpful knowledge remains a challenge in the WEBBRAIN task. (2) ReGen surpasses the other three retrievers on most metrics, which implies that ReGen could retrieve more helpful knowledge. With

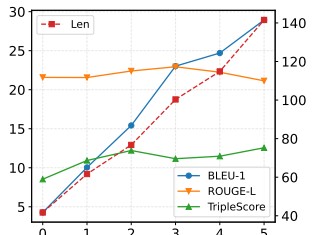

Figure 2: Performance of ReGen (Large) with different numbers of retrieved references.

such knowledge, ReGen could generate better answers regarding factuality, confirming the effectiveness of the fact-enhancement strategies applied in ReGen. (3) Regarding retrieval performance, our Regen outperforms all baselines on all metrics (see full retrieval results in Table 9).

We also evaluate ReGen's performance with different numbers of retrieved references. Specifically, we randomly sample 100 queries from the test set and feed them into ReGen's generator together with zero to five references retrieved by ReGen's retriever. Figure 2 illustrates the results. Generally, with more references, BLEU score and text length increase, but ROUGE score and triple score fluctuate. This implies that more content is generated, but the quality cannot be guaranteed. The ROUGE score achieves the best with three references. Combined with the retriever's performance, we attribute this effect to the noisy nature of retrieved references. Finally, we note that, even with references, generating long and faithful text is still a challenging problem.

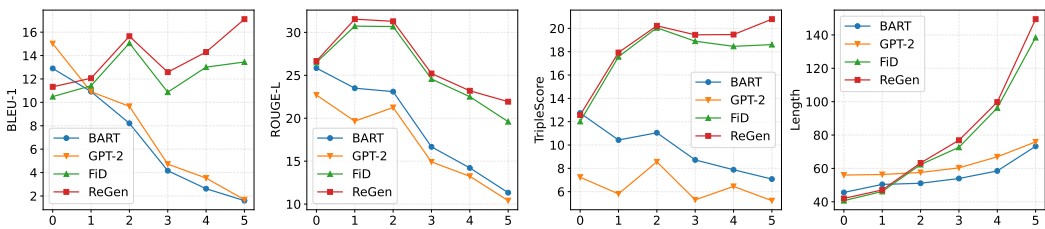

Figure 3: Performance of different models with different numbers of ground-truth references.

**Impact of Number of References**    As references provide evidence for generating factual content, their quantity will impact the quality of generation. To investigate this, we randomly sample 500 queries from the test set with zero to five references, respectively (3,000 in total). Different from the previous analysis, we use ground-truth references in this experiment. Figure 3 shows the results. We observe: (1) For models using no reference (*i.e.*, BART and GPT-2), their performance is decreasing ($0 \rightarrow 5$ references). Intuitively, target texts with more references are much more difficult to generate, as they require more knowledge to construct the content. This result validates our assumption that writing human-like descriptions for queries heavily relies on external knowledge, which is difficult for pre-trained models to store as parameters. (2) The performance of FiD and ReGen are improving in terms of BLEU and TripleScore. This is because target texts with more references are usually longer, and the references can provide more information for generation. However, as suggested by ROUGE, more noise also appears in the generated text.

**Performance vs. GPT-3**    GPT-3 (Brown et al., 2020) is a large-scale pre-trained language model with 175B parameters. The model demonstrates strong few- or zero-shot performance on many text-based tasks, as vast knowledge has been stored in its parameters. To compare this strong model with ours, we first randomly sample 100 queries from the test set. Then, for GPT-3, we use the prompt "`Introduce [X] in Wikipedia-style.`" to generate zero-shot results.[6] For our model, we use the query and evidence provided by the retriever as input. The results of automatic metrics and human annotation are shown in Figure 4 and the last row of Table 5. Our model outperforms GPT-3 by a large

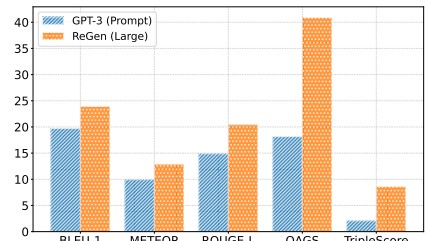

Figure 4: Performance of GPT-3 (Prompt) and our ReGen (Large) on 100 test samples.

margin in terms of BLEU, METEOR, and ROUGE. This implies that our model can generate texts that are closer to the ground-truth (written by humans), indicating that explicit knowledge in web corpus is essential for generating natural and informative text. Only increasing the size of the language model would fail to capture such knowledge. We also observe a clear improvement in our model on factualness metrics. Even with a large number of parameters, it is evident that GPT-3 cannot guarantee factualness. In contrast, evidence from Web corpus can reduce these hallucinations. Due to the limited space, please refer to **Appendix** for more experimental results and discussion.

# 6    CONCLUSION

In this paper, we introduce a new NLP task, WEBRAIN, which aims to generate a fluent, informative, and factually correct description for factual queries via mining supporting evidence from a large web corpus. We construct a large-scale dataset WebBrain-raw as the data foundation of WEBRAIN. WebBrain-raw is constructed by crawling all Wikipedia articles and the corresponding reference articles. Through empirical analysis, we found that most current NLP techniques lack the ability to maintain the factualness of the WEBRAIN task. To improve the performance on this task, we propose a new framework ReGen which uses several specifically designed strategies. It outperforms all baselines on automatic and human evaluation. We believe that WEBRAIN would highlight a valuable research pathway in which AI models could acquire knowledge from the Web world by themselves and serve human beings by fulfilling a broader range of fact-oriented information needs.

---

[6]We use the model `text-davinci-002` provided in `https://beta.openai.com/playground`.

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

## A    Implementation Details

In our ReGen, the retriever is training on the WebBrain-R dataset for five epochs, initializing from `efficient-splade-V-large`[7]. We set the batch size to 192, the max query length to 16, and the max passage length to 256. We use the AdamW optimizer with a learning rate of 2e-5. We set $(\lambda_q, \lambda_d)$=(5e-4, 5e-3) for the Eq. 6. For baseline models, we train SPLADE and DPR with their official settings on the WebBrain-R dataset for five epochs. The underlying RetroMAE of DPR is initialized from the BEIR checkpoint [8]. We implement the models with Pytorch and Pytorch-Lightning [9]. We train the models on 8 Tesla V100 32G GPUs. For the BM25 retriever, we use the Pyserini tool with its default parameter setting (Lin et al., 2021). As for the generator, we use BART-based and BART-large checkpoints provided by HuggingFace (Wolf et al., 2019) to initialize the model. For training, we use 32 and 64 Tesla V100 32G GPUs for the base and large models, respectively. Correspondingly, the batch size of each GPU is set as 8 and 4. AdamW optimizer (Loshchilov & Hutter, 2019) is used for optimization with a learning rate of 5e-5. In the inference stage, we apply beam search (width=5), and the maximum generation length is set as 512. This generation setting is also used for GPT-3 (prompt). We provide our code in an anonymous repository for review.[10]

**Construction of WebBrain-R**    For each reference document, we perform reference passage selection: the passage with the highest $P_{ST}$ value is kept. The reason for the passage selection is to keep the working dataset at a reasonable size (considering efficiency), and meanwhile to contain at least one relevant passage to support generation. In this way, as shown in Table 2, we construct a million-level corpus for WebBrain-R. In this corpus, the Wiki title and the selected passages form a positive pair. To obtain negative pairs, we index the corpus with a BM25 engine (Lin et al., 2021), which mines negative passages for each query by randomly selecting four passages from the top 30-50 results.

**Construction of WebBrain-G**    Similar to the construction of WebBrain-R, we perform reference passage selection for each reference document. Then, for each sample, we concatenate the Wiki title and each reference passage respectively as input. The first section of Wiki page is used as target.

## B    Human Evaluation

To further evaluate our ReGen model, we perform human evaluation in terms of three metrics: (1) **Fluency** evaluates the linguistic coherency of the generated text; (2) **Informativeness** measures whether the generated text gives a response in specific details based on the query and references; and (3) **Faithfulness** reflects the proportion of the generated text that can be supported by the retrieved supporting evidence. To reduce the influence of objective factors in the manual annotation, we perform a pairwise evaluation method: by giving the results generated by ReGen and other baseline models, three annotators are asked to judge which text is better or if there is a tie. For fairness, the results from different models are randomly shuffled, and the annotators would not know the underlying model in advance. The evaluation results are shown in Table 5.

## C    Reference Mark Correction

A reference mark is used to indicate the source of a referenced text. Correct reference marks can directly improve the factualness and readability of the generated text. However, as the FiD model simultaneously performs attention on the representations of all tokens in multiple references, it is hard to identify the correct source of a generated sentence. We try to alleviate this problem by using a **Reference mark-Enhanced Decoding** (RED) strategy: in the inference stage, if a reference mark (other than `[0]`) has the largest generation probability at a decoding step, we extract the last generated sentence and match it with each reference. Then, the reference with the largest word overlap

---

[7] https://huggingface.co/naver/efficient-splade-V-large-query and https://huggingface.co/naver/efficient-splade-V-large-doc

[8] https://huggingface.co/SamuelYang/SentMAE_BEIR

[9] https://pytorch-lightning.readthedocs.io/en/stable/

[10] https://anonymous.4open.science/r/WebBrain/

Table 5: Human annotation results. The first three pairs are all base models.

| Comparison | Fluency | | | Informativeness | | | Faithfulness | | |
|---|---|---|---|---|---|---|---|---|---|
| | Win | Tie | Lose | Win | Tie | Lose | Win | Tie | Lose |
| ReGen vs. BART | 39% | 52% | 9% | 29% | 63% | 8% | 42% | 46% | 12% |
| ReGen vs. GPT-2 | 42% | 43% | 15% | 30% | 59% | 11% | 54% | 30% | 16% |
| ReGen vs. FiD+BART | 22% | 63% | 15% | 17% | 74% | 9% | 37% | 51% | 12% |
| ReGen (Large) vs. GPT-3 (Prompt) | 48% | 22% | 30% | 34% | 51% | 15% | 72% | 4% | 24% |

Table 6: Performance of ReGen (Large) on samples with different numbers of retrieved references.

| #Refs. | BLEU-1 | BLEU-4 | METEOR | ROUGE-L | CIDEr | QAGS | TripleScore | Len |
|---|---|---|---|---|---|---|---|---|
| 0 | 4.27 | 1.48 | 7.67 | 21.58 | 46.99 | 17.14 | 8.53 | 41.71 |
| 1 | 10.04 | 3.17 | 9.00 | 21.56 | 26.85 | 53.46 | 10.93 | 61.65 |
| 2 | 15.42 | 5.23 | 10.81 | 22.40 | 21.14 | 49.04 | 12.21 | 76.73 |
| 3 | 23.02 | 7.34 | 13.21 | 22.94 | 19.08 | 49.36 | 11.16 | 100.32 |
| 4 | 24.70 | 7.72 | 13.64 | 22.24 | 19.30 | 49.46 | 11.48 | 114.86 |
| 5 | 28.94 | 8.74 | 14.87 | 21.27 | 19.30 | 48.72 | 12.56 | 141.55 |

Table 7: Performance of models on samples with different numbers of ground-truth references.

| Model | #Refs. | BLEU-1 | BLEU-4 | METEOR | ROUGE-L | CIDEr | QAGS | TripleScore | Len |
|---|---|---|---|---|---|---|---|---|---|
| BART | 0 | 12.90 | 4.87 | 11.28 | 25.84 | 49.31 | 25.22 | 12.75 | 45.60 |
| | 1 | 10.92 | 4.11 | 10.04 | 23.50 | 39.45 | 29.40 | 10.43 | 50.42 |
| | 2 | 8.21 | 3.59 | 10.11 | 23.10 | 40.13 | 30.70 | 11.05 | 51.10 |
| | 3 | 4.16 | 1.20 | 7.39 | 16.67 | 13.16 | 27.75 | 8.72 | 53.96 |
| | 4 | 2.62 | 1.08 | 7.20 | 14.21 | 10.91 | 25.03 | 7.88 | 58.49 |
| | 5 | 1.60 | 0.41 | 6.13 | 11.33 | 1.96 | 25.22 | 7.08 | 73.29 |
| GPT-2 | 0 | 15.02 | 4.11 | 10.34 | 22.70 | 27.94 | 23.88 | 7.24 | 56.01 |
| | 1 | 10.91 | 2.96 | 8.75 | 19.66 | 18.84 | 25.32 | 5.81 | 56.35 |
| | 2 | 9.66 | 3.80 | 9.84 | 21.25 | 33.77 | 30.09 | 8.56 | 57.57 |
| | 3 | 4.72 | 1.19 | 6.70 | 14.91 | 8.99 | 25.89 | 5.30 | 60.34 |
| | 4 | 3.53 | 1.29 | 6.82 | 13.24 | 10.36 | 22.89 | 6.45 | 66.95 |
| | 5 | 1.68 | 0.33 | 5.60 | 10.40 | 1.49 | 23.08 | 5.23 | 75.90 |
| FiD | 0 | 10.49 | 4.18 | 10.84 | 26.50 | 49.18 | 20.68 | 12.02 | 40.68 |
| | 1 | 11.42 | 5.16 | 13.06 | 30.73 | 70.52 | 59.63 | 17.57 | 46.17 |
| | 2 | 15.06 | 7.18 | 15.51 | 30.68 | 66.91 | 57.91 | 20.05 | 62.31 |
| | 3 | 10.87 | 4.04 | 13.48 | 24.57 | 29.23 | 59.46 | 18.90 | 72.64 |
| | 4 | 13.01 | 5.58 | 14.96 | 22.51 | 17.81 | 55.26 | 18.46 | 96.29 |
| | 5 | 13.45 | 4.97 | 14.36 | 19.60 | 14.30 | 50.64 | 18.61 | 138.41 |
| ReGen | 0 | 11.33 | 4.39 | 11.11 | 26.66 | 50.04 | 19.62 | 12.57 | 42.02 |
| | 1 | 12.06 | 5.67 | 13.36 | 31.54 | 76.35 | 59.05 | 17.93 | 47.10 |
| | 2 | 15.65 | 7.76 | 15.78 | 31.29 | 67.90 | 59.39 | 20.22 | 63.27 |
| | 3 | 12.58 | 4.84 | 14.09 | 25.21 | 36.29 | 58.84 | 19.45 | 76.87 |
| | 4 | 14.30 | 6.17 | 6.17 | 23.20 | 20.48 | 55.13 | 19.47 | 99.75 |
| | 5 | 17.11 | 7.24 | 15.95 | 21.93 | 19.33 | 50.60 | 20.79 | 149.45 |

ratio is selected as the source of the sentence (*i.e.*, its reference mark is used for this sentence). To test this strategy, for each generated text, we compute a `refer score` as follows:

$$P_r = \frac{1}{n} \sum \frac{|\mathcal{T} \cap \mathcal{V} - S|}{|\mathcal{V} - \mathcal{S}|}, \tag{12}$$

where $n$ is the number of sentences with references; $\mathcal{T}$ is the set of words in the sentence; $\mathcal{V}$ is the set of words in the corresponding reference; and $\mathcal{S}$ is the set of stopwords. This score is similar to the STP score defined in Eq. 3 but focuses on how much of the content of the sentence is quoted from the corresponding reference. From the results shown in Figure 5, we can see this strategy improves refer scores greatly but has less influence on other metrics. This indicates that the reference mark

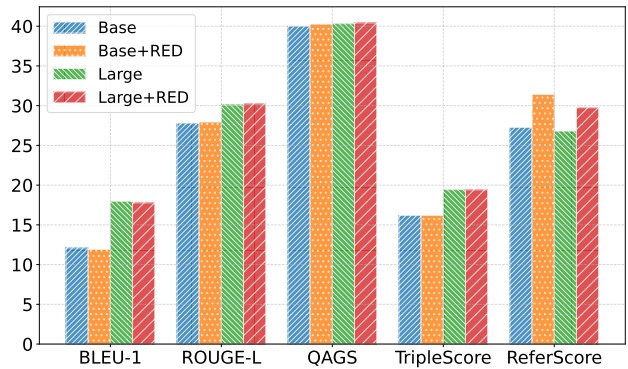

Figure 5: Influence of reference mark-enhanced decoding.

Table 8: Performance of ReGen (Large) with different retrievers on the full and the selected passage set.

| Model | BLEU-1 | METEOR | ROUGE-L | CIDEr | TripleScore |
|---|---|---|---|---|---|
| ReGen (Full) | **27.81** | 14.89 | **22.00** | 19.40 | **13.68** |
| + BM25 | 27.16 | 14.54 | 21.38 | 17.32 | 12.98 |
| + DPR | 24.57 | 13.17 | 19.82 | 15.21 | 10.74 |
| + SPLADE | 25.60 | **14.98** | 20.71 | 14.24 | 12.35 |
| ReGen (Selected) | 24.71 | 12.99 | 21.19 | **20.12** | 10.97 |
| + BM25 | 25.76 | 13.73 | 19.94 | 14.85 | 10.43 |
| + DPR | 20.42 | 12.71 | 16.11 | 6.44 | 6.05 |
| + SPLADE | 23.84 | 14.24 | 18.98 | 10.80 | 9.48 |
| Oracle | 17.97 | 15.31 | 30.09 | 62.39 | 19.43 |

is labeled more accurately. We apply this strategy in our demo (described in Appendix H), which also improves the demo's readability. However, our strategy is just an early exploration of reference mark accuracy. It is only involved in the inference stage, but not in model training. Moreover, how to evaluate the reference mark's accuracy is also an unsolved problem. We leave these problems for our future work.

## D MOTIVATION AND IMPACT OF REFERENCE PASSAGE SELECTION

As introduced in Section 3.2, during the dataset construction, we perform reference passage selection to build both WebBrain-R and WebBrain-G. They have different purposes.

For WebBrain-R, we construct a selected reference passage corpus for retriever training and evaluation. The reason is that WebBrain-R is considered a working dataset for training, on which the experiments should take a reasonable amount of time to ensure efficiency. Thus, instead of using the full reference passage set, which requires a long time to index, we keep the selected passage corpus at a reasonable size (million level) to evaluate the retrievers in the training phase.

For WebBrain-G, our passage selection strategy can ensure that the provided references are relevant (can be treated as *oracle* references), thus helping the model learn to extract useful information from the references. Such filtering may not be possible in practice, but it is a common approach used to build a workable dataset in many previous studies.

To simulate real-world scenarios, we conduct an end-to-end evaluation in Section 5.3. In this experiment, ReGen's retriever needs to find evidence from the full reference passage set (containing about 204 million passages, which is significantly larger than WebBrain-R), and ReGen's generator has to generate texts based on the retrieved evidence. As a comparison, we also conduct an experiment

Table 9: Performance of various retrievers on WebBrain-R.

| Model | R@1 | P@1 | R@5 | R@10 | R@20 | MRR | MAP |
|---|---|---|---|---|---|---|---|
| BM25 | 0.290 | 0.292 | 0.558 | 0.645 | 0.716 | 0.411 | 0.410 |
| DPR (BERT) | 0.192 | 0.193 | 0.348 | 0.403 | 0.457 | 0.263 | 0.263 |
| DPR (RetroMAE) | 0.228 | 0.229 | 0.390 | 0.438 | 0.478 | 0.301 | 0.301 |
| SPLADE | 0.389 | 0.392 | 0.724 | 0.815 | 0.877 | 0.537 | 0.536 |
| ReGen | **0.399** | **0.401** | **0.746** | **0.839** | **0.896** | **0.551** | **0.550** |

Table 10: Performance of different models on WebBrain-G. The models are evaluated by semantic similarity metrics. The best results are in bold.

| | BERTScore | | | BARTScore |
|---|---|---|---|---|
| Model | P | R | F1 | (log-likelihood) |
| BART (Base) | 96.8 | 94.4 | 95.5 | -4.375 |
| BART (Large) | 97.6 | 94.8 | 96.2 | -4.280 |
| GPT-2 | 87.3 | 85.4 | 86.3 | -4.560 |
| FiD+BART (Base) | 96.2 | 93.9 | 95.1 | -4.091 |
| FiD+BART (Large) | 97.6 | 94.9 | 96.2 | -4.014 |
| ReGen (Base) | 96.2 | 93.9 | 95.0 | -4.071 |
| ReGen (Large) | **97.7** | **97.8** | **97.8** | **-3.936** |
| ReGen (w/ Retriever) | 90.2 | 84.5 | 87.3 | **-4.303** |
| + BM25 | **91.6** | 91.3 | **91.2** | -4.326 |
| + DPR | 83.3 | **91.4** | 87.1 | -4.664 |
| + SPLADE | 88.1 | 85.5 | 86.8 | -4.381 |

on WebBrain-R. This helps us understand how the size of the passage set impacts the end-to-end performance of ReGen. The results are shown in Table 8. We find that:

(1) When using the selected reference passage set, all retrievers' performance decreases over almost all metrics, indicating that the retrievers can find more useful information that benefits generation in the full passage set.

(2) When using the full passage set, our ReGen's retriever surpasses BM25 regarding BLEU-1 and METEOR, proving that ReGen's retriever generalizes better in the large-scale passage set, and therefore finds passages with better informativeness.

(3) Though the full passage set is much larger, the performance of all the retrievers has a similar tendency, which verifies that reference passage selection is a simple yet effective method for dataset construction. Evaluating models' performance on the selected passage set can well reflect its performance on the full passage set, which is an important finding that allows researchers to conduct experiments on the WebBrain-R.

(4) Although ReGen's generator is trained with oracle passages, it can generalize well to use the retrieved passages for generation. This demonstrates that learning on WebBrain-G can help the model learn to extract useful information from the references.

We will also release the full passage set to support in-depth research on the WEBBRAIN task.

## E EVALUATION OF SEMANTIC SIMILARITY

In Section 5.2, we evaluate models by several $n$-gram overlapping-based metrics. Recently, there have been some new model-based metrics that can evaluate text similarity at the semantic-level. Therefore, we also employ BERTScore (Zhang et al., 2020) and BARTScore (Yuan et al., 2021) for evaluation.[11] For BERTScore, we report precision (P), recall (R), and F1. For BARTScore,

---

[11]We use the checkpoint provided in their Github repository. BERTScore: `https://github.com/Tiiiger/bert_score`. BARTScore: `https://github.com/neulab/BARTScore`.

Table 11: Performance of GPT-3 (Prompt) and our ReGen (Large) on 100 test samples. The first prompt is "Introduce [X]:", while the second prompt is "Introduce [X] in Wikipedia-style.". The second prompt has a hint of generating Wikipedia-like content.

| Model | BLEU-1 | BLEU-4 | METEOR | ROUGE-L | CIDEr | QAGS | TripleScore |
|---|---|---|---|---|---|---|---|
| GPT-3 (Prompt 1) | 17.24 | 2.19 | 9.14 | 13.57 | 6.70 | 17.00 | 0.29 |
| GPT-3 (Prompt 2) | 19.72 | 3.60 | 9.93 | 14.89 | 9.33 | 18.16 | 2.08 |
| ReGen (Large) | 28.94 | 8.74 | 14.87 | 21.27 | 19.30 | 48.72 | 12.56 |

it is an average log-likelihood, so its value is negative. Higher scores indicate that the generated text is more semantically similar to the ground-truth text. The results are shown in Table 10. In the upper side of the table, we can observe that our ReGen (large) achieved the best results among all models. This is consistent with the results in Table 3, demonstrating the superiority of ReGen in generating semantically correct text. More concretely, the recall score of the ReGen (large) is considerably higher than that of other models. This indicates that it can cover the content of ground-truth texts more effectively. In the lower side, we compare the performance of ReGen (large) with other retrievers. Considering BERTScore, the recall score of DPR (a dense retriever) is higher than precision score, showing the opposite trend to other retrievers. This suggests that the retrieved results returned by DPR can cover more content with the ground-truth target but also contain much noise. Combining references from different retrievers may be a promising strategy for improving generation quality.

## F   ADDITIONAL PERFORMANCE COMPARISON WITH GPT-3

In Section 5.3, we compare our ReGen model with GPT-3. As GPT-3 is tested by prompt, the selection of the prompt template will directly influence its performance. In our experiments, we consider two different kinds of templates, depending on whether or not they explicitly specify to generate text in Wikipedia-style. After evaluating and comparing the generation quality, we finally choose "Introduce [X]:" and "Introduce [x] in Wikipedia-style." as the prompt templates. The results are shown in Table 11. From the results of GPT-3 with different prompts, we can confirm that a prompt for generating Wikipedia-style text can considerably improve the naturalness and factualness of text. This also indicates the importance of prompt engineering when employing large-scale pre-trained language models. Moreover, ReGen can generate texts that are both closer to the ground-truth (achieving higher BLEU, METEOR, ROUGE, and CIDEr) and more factual (obtaining higher QAGS and TripleScore). This confirms that our use of explicit knowledge in Web corpus is essential for generating natural, informative, and factual text.

## G   CASE STUDY

We perform a qualitative analysis of our model based on a case study. Specifically, we first compare the generated results of different models on queries with or without references. Then, we show a series of generated results for a specific query with different numbers of references.

**Comparison of Different Models**   We first study the generation of queries without references. As shown in Table 12, where no reference is given, FiD, BART, and ReGen generate similar results, but ReGen can provide more details. We attribute this to the warm-up strategy, which can effectively mitigate the gap of tasks between BART's pre-training and our fine-tuning. Besides, it is interesting to see that GPT-3 can also generate correct information about the election date and post. We relate this to the large number of parameters in GPT-3. They are capable of storing large amounts of knowledge. However, GPT-3 also generates some detailed information about the candidates that is factually incorrect (*i.e.*, hallucination). In the second case shown in Table 13, we use green and red colors to label the correct and incorrect information. We can see with one reference provided, both FiD and ReGen can generate more faithful text than BART. This reflects the importance of external knowledge. Furthermore, we can also observe that ReGen can better capture clues from the reference. The result also demonstrates the effectiveness of our proposed warm-up strategy, as it can facilitate our model's capability of sentence-reference matching. However, GPT-3 performs poorly

Table 12: Generated results of a query without reference. The green part of information is only correctly generated by ReGen.

---

**Query**: 2016 Pateros local elections
**Ground-truth reference**: None
**Ground-truth target**: Local elections in pateros were held on may 9, 2016 within the philippine general election. The voters elected for the elective local posts in the city: the mayor, vice mayor, district representative, and councilors, six in each of the city's two legislative districts.
**BART (Large)**: Local local elections were held in pateros on may 9, 2016 within the philippine general election. The voters elected for the elective local posts in the city: the mayor, vice mayor, and ten councilors.
**FiD (Large)**: Local elections was held in pateros on may 9, 2016 within the philippine general election. The voters elected for the elective local posts in the city: the mayor, vice mayor, and eight councilors.
**GPT-3**: Pateros held its 2016 local elections on May 9, 2016. Elections were held for the mayor, vice mayor and councilors. incumbent Mayor Jaime Medina was term-limited and could not run for re-election. Medina was first elected in 2004 and was re-elected in 2007 and 2010. incumbent Vice Mayor Nestor Padilla was term-limited and could not run for re-election. Councilor Mario Marasigan was the only incumbent councilor who ran for re-election. A total of 15 candidates ran for mayor, while 24 candidates ran for vice mayor. 55 candidates ran for councilor.
**ReGen (Large)**: Local elections was held in pateros on may 9, 2016 within the philippine general election. The voters elected for the elective local posts in the city: the mayor, vice mayor, the two congressmen, and the councilors, six in each of the city's two legislative districts.

---

in this case. It can generate a natural and fluent text with lots of information, but most of the facts are incorrect. Due to the fact that all knowledge is stored as parameters in GPT-3, it is challenging to determine which part is relevant to the current query. In contrast, ReGen can leverage the knowledge provided in given references, thus generating more factual text.

**Impact of Number of References**   We also evaluate the generated results by feeding various numbers of references into ReGen. Here, we show a query selected from the test set used in the section "Impact of Retrievr". All used references are retrieved by ReGen's retriever. The input query and retrieved references are presented in Table 14, while the generated results are shown in Table 15. For the generated results with $n$ references, we use the first $n$ retrieved references in generation. Generally, we can observe that the generated results become longer when more references are given, reflecting the importance of references in content construction. Then, even without reference, Re-Gen is able to generate text with some factual knowledge. This suggests that ReGen can also store some knowledge in its parameters. We can also observe that the text factualness is greatly enhanced when more references are provided. Nevertheless, we still notice some problems in both the retrieved references and the generated results. As for the references, some of them are uninformative (such as the second reference) or less relevant (such as the fifth reference). They introduce noise to the generation process. As for the generated results, they contain some hallucinations caused by mixing information from various references. All of these results demonstrate that there is much room for improvement in existing methods.

## H   SYSTEM DEMONSTRATION

Though there is a gap between our study and our ultimate goal of generating Wikipedia articles, we still explore potential application scenarios for our current model. We build a demonstration system as shown in Figure 6, Figure 7, and Figure 8. This system can generate either a short article or a full page. For generating short articles, we test both entity-style queries similar to Wikipedia titles and natural language-style queries. We illustrate and discuss them respectively.

As shown in Figure 6, we first input an entity-style query of "Google Translate". This scenario is the same as model inference. We can see our system can generate content relevant to the query. Besides, with the reference mark-enhanced decoding strategy, we can see the reference mark is generated more accurately, which greatly enhances the readability of the generated text.

In addition, we also examine the system with natural language-style queries (shown in Figure 7). Intriguingly, we find that though ReGen is trained on entity query-based dataset, it can generalize well in some natural language queries, which have more specific information-seeking intent. Com-

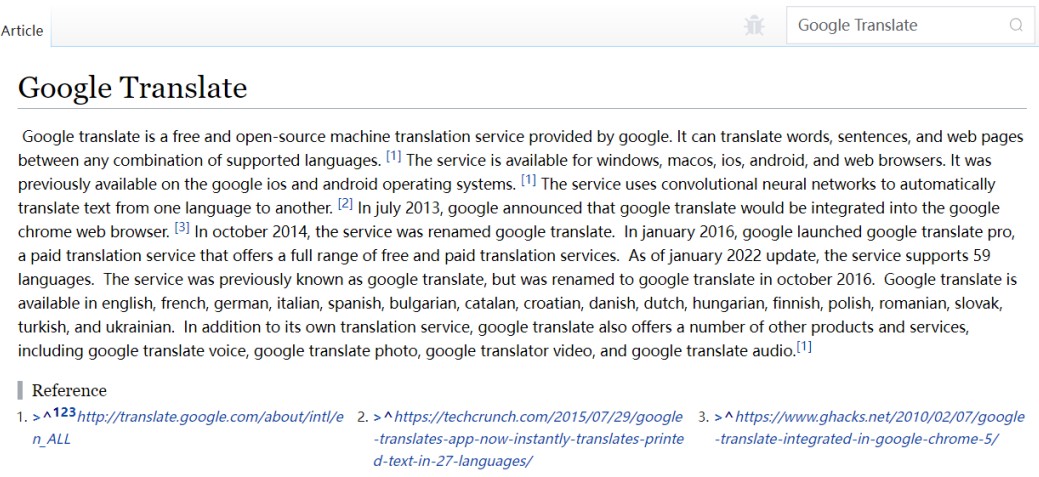

Figure 6: Screen shot of our system: entity query.

pared to the query "Google Translate", the intent of the query "how Google Translate work" has a more specific intent. As a result, the generated text is closer to the intrinsic functionality of Google Translate.

We also try to generate a full Wikipedia-style page with a heuristic method. Specifically, we first employ another model to generate some section titles for the input query. Then, we respectively concatenate the input query and each generated section name as new queries. Finally, ReGen can generate a short article for the original query and each section recursively. Figure 8 demonstrates the result of "Google Translate". The generated section titles are "early years of Google Translate", "neural translation model", and "translate tools". Through the case, we can find: though the generated text can provide a good introduction to the section title, different section titles do not share a reasonable taxonomy given the input query. For example, the section title "translate tool" seems to be more general than "Google Translate". We will continue exploring how to generate full Wikipedia pages in our future work.

## I  LIMITATION AND FUTURE WORK

In this work, we explore generating short factual articles (like Wikipedia pages) for queries by collecting evidence from the Web. This is a very early step towards our ultimate goal: automatically building Wikipedia pages for unseen topics. We collect a new dataset to support future research, evaluate several existing models on the new task, and propose some effective strategies. However, there are still many limitations in our study that need further investigation in the future.

(1) The retriever is far from perfect. As shown in our experiments, ReGen's performance with retrieved evidence is worse than that with oracle references. So, two problems appear: (a) how to improve the retriever's performance; and (b) how to generate a better result with non-perfect references.

(2) As we use FiD as our backbone model, it is hard to determine the reference mark accurately. Moreover, the reference mark accuracy cannot be measured by existing metrics. We plan to design new attention mechanisms to facilitate the model's judgment on these reference marks.

(3) In this study, we try to generate a short article for given queries. However, the original Wikipedia page contains more information than a short article. Generating long articles will be another challenge in the future.

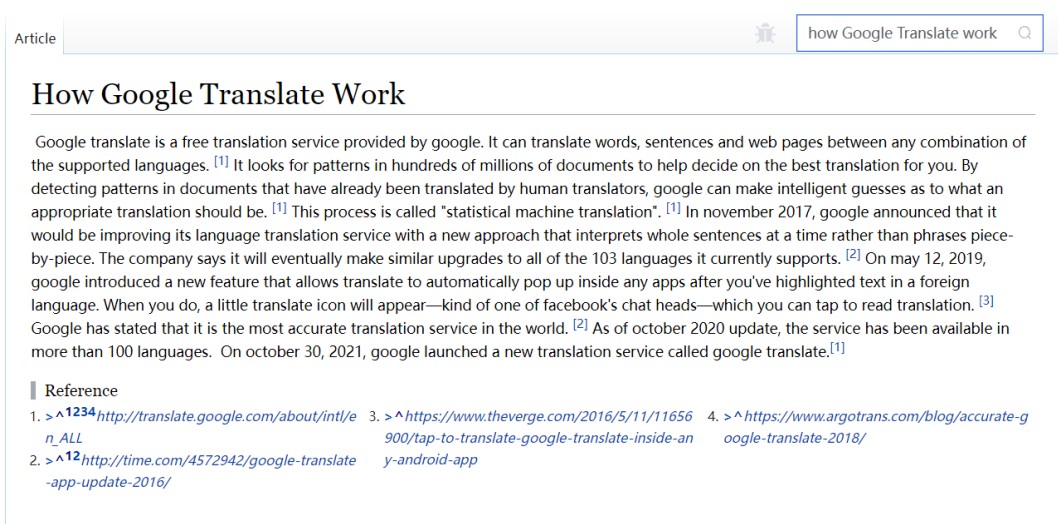

Figure 7: Screen shot of our system: natural language query.

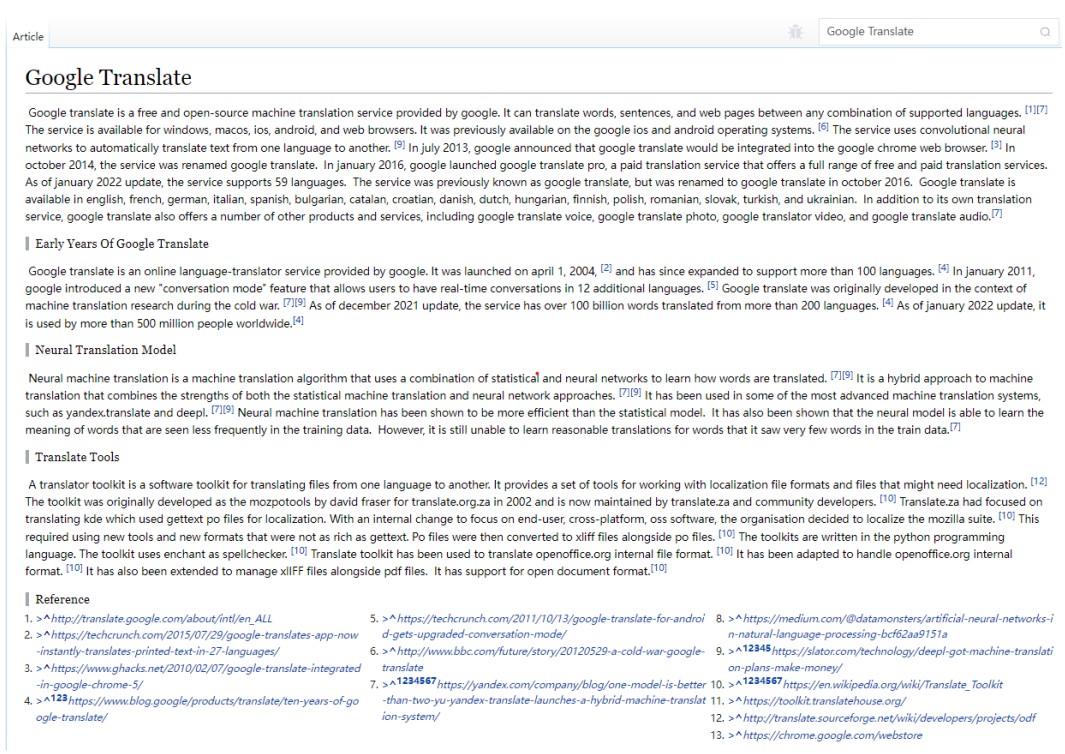

Figure 8: Screen shot of our system: entity query for generating a full Wikipedia page.

Table 13: Generated results of a query with reference. The green texts are correct information, while the red ones are incorrect.

---

**Query**: Charles Lovell (trade unionist)

**Ground-truth reference 1**: The Guardian Charles Lovell, who has died aged 91, led the plumbers' trade union the ptu (later the eetpu) from the 1960s to the 1980s, a period of great turmoil and change in the world of industrial relations. Charles developed partnerships between his union and employer bodies representing the industries in which his members served. The outcome was enormously beneficial to both parties and to the industry. Son of Michael, a warehouseman, and Isabella, Charles was born in Newcastle upon tyne and began work in 1939. He was offered an apprenticeship as a plumber and joined the plumbing trade union, but as the second world war approached, he was drafted to swan hunter's shipyard under an essential works order. He was never happy there and returned to construction as soon as he was able. After the war, he also immersed himself in trade union activities and became chairman of the wallsend trades council and a labour councillor, as well as learning esperanto and providing for his beloved wife, Irene (nee lamb), whom he married in 1944, and their three daughters and two sons. In 1966 he was elected to the executive of the ptu and in 1968 became its general secretary. To make his union more secure, one of his first acts was to merge it with the electrical trade union to form the eetpu.

**Ground-truth target**: Charles Lovell (1923 – 13 october 2014) was a british trade unionist. Born in Newcastle-upon-Tyne, Lovell left school in 1939 and began an apprenticeship as a plumber, also joining the plumbing trades union. Soon after, he was drafted to work at the swan hunter shipyard, but returned to plumbing immediately after world war ii.[1] Lovell joined the labour party and was elected as a councillor, and also chaired wallsend trades council. He was elected to his union's executive council in 1966, and two years later won election as its general secretary.[1] He immediately led it into a merger with the electrical trades union, becoming national officer of the new electrical, electronic, telecommunications and plumbing union. In this role, he created a national pension scheme for plumbers, and championed more flexible apprenticeships, which could be completed under a variety of employers.[1] Lovell retired in 1987, and was made an officer of the order of the British empire. In his spare time, he learned Esperanto.[1]

**BART (Large)**: Charles Lovell was a British trade unionist. Born in london, Lovell worked as a coal miner, and became active in the miners' federation of great Britain. He was elected as the union's general secretary, and also served on the general council of the national union of mineworkers. Lovell also served as a member of the executive of the trades union congress, and was a delegate to the general conference of the international labour organization. He stood unsuccessfully for the labour party at the 1918 united kingdom general election, but was elected at the 1922 general election. In 1923, he was appointed as general secretary of the union, a post he held until his death.

**FiD (Large)**: Charles Lovell (1921 – 30 June 2011) was a British trade unionist. Born in Newcastle upon Tyne, Lovell began working as a plumber in 1939. He joined the plumbing trade union, and was elected to its executive in 1953. He was also elected to the wallsend trades council, and served as its chairman in the 1960s.[1] In 1966, Lovell was elected as the general secretary of the plumbers' trade union, and led the union through a period of turmoil and change in the world of industrial relations.[1] Lovell was also active in the labour party, serving as a councillor on wallsend borough council and as a member of the national executive committee of the trades union congress.[1] He retired from the union in 1986, and died in 2011, aged 91.[1]

**GPT-3**: Charles Lovell was a noted trade unionist in the early 20th century. He was born in England in 1868, and began working in the coal mines at the age of 10. He later moved to the United States, where he worked in the coal mines of Pennsylvania. He became active in the United Mine Workers of America (UMW), and was elected to the union's international executive board in 1902. He also served as the UMW's vice president from 1909 to 1913. Lovell was a leading figure in the union's fight for better working conditions and higher wages for miners. He played a key role in the UMW's successful strike against the coal companies in 1902. He also helped to organize the union's successful strike against the anthracite coal companies in 1905. Lovell was a leading advocate of industrial unionism, and played a key role in the formation of the Industrial Workers of the World (IWW) in 1905. He served as the IWW's first general secretary from 1905 to 1908. Lovell's radical views and activities caused him to be blacklisted by the coal companies, and he was unable to find work in the coal mines after 1908. He later found work as a journalist and editor for several leftist publications. He died in New York City in 1932.

**ReGen (Large)**: Charles Lovell (1928 – 15 July 2016) was a British trade unionist. Born in Newcastle upon Tyne, Lovell began work in 1939. He joined the plumbing trade union, but as the second world war approached, he was drafted to swan hunter's shipyard under an essential works order. He was never happy there and returned to construction as soon as he was able.[1] After the war, he also immersed himself in trade union activities and became chairman of the wallsend trades council and a labour councillor, as well as learning Esperanto and providing for his beloved wife, Irene (nee lamb), whom he married In 1944, and their three daughters and two sons. In 1966 he was elected to the general council of the trades union congress.[1] He led the ptu from the 1960s to the 1980s, a period of great turmoil and change in the world of industrial relations. Charles developed partnerships between his union and employer bodies representing the industries in which his members served.[1] The outcome was enormously beneficial to both parties and to the industry.[1]

Table 14: A query with five retrieved references. The ground-truth target and generated results are shown in the next table.

**Query**: Makerere University School of Law

**Retrieved reference 1**: "MUK Students Challenge Proposal on Tuition Increment" Makerere suggested and approved a 10% tuition increase for all their courses starting with the 2014/2015 intake. A 10% increment would mean students at the school of law that have been paying 1.2 million shillings per semester would pay an extra 120,000 shillings. The school of medicine where tuition has been in the range of 1.4 to 1.6 million shillings per semester would have a 140,000 and 160,000 Uganda shillings extra fee. The fees increment has been justified by the university management as a means of raising the internal revenue of the university to cater for the burgeoning expenses like lecturers' salaries and research fees. Makerere university relies on three sources of funding; student's tuition, government contribution and donor funds to run its 122 billion shillings annual budget. Student's contribution accounts for over 70% of the budget. Ivan Bwowe, The university guild president told journalists that they were concerned that the university is not looking at alternative sources of funding and only relying on contributions from students' tuition. He says the students are not prepared to let the 10% tuition increment prevail. The university chancellor professor Mondo Kagonyera on Friday decried the culture of strikes in Makerere. He said he would not negotiate with students if they continue to use violence to hold the university at ransom. Despite an earlier agreement with the vice chancellor that food will be left in the hands of a private contractor, the students still raised queries over how the process would be handled in their petition.

**Retrieved reference 2**: "List of Universities in Uganda Accredited To Teach Law" Uganda law council +256 414341673 info @ lawcouncil.co.ug georgian house, Kampala-Uganda the law council committee on legal education & training accredited the universities listed below to teach the law degree programmes

**Retrieved reference 3**: BBC.com a historian of east Africa, Derek Peterson, says the fire is a disaster for Uganda and for east Africa. The main building held student records and the archives of the Makerere University "the building holds student records, and the basement is full of archive files spanning the whole history of the institution," he tweeted, adding that he had been intending to help organise a project to catalogue the collection. In 1970 , African presidents, including Makerere graduate Julius Nyerere of Tanzania, attended a ceremony at the university. Makerere was first established in 1922 as a technical school and has grown into a widely respected university. The construction of the "ivory tower" building began in the 1930s and was completed in 1941 its alumni include independence-era leaders such as Julius Nyerere of Tanzania, renowned writers including Kenya's Ngugi wa Thiong'o, academics and clergy like John Sentamu, the recently retired Anglican archbishop of York. Makerere university is Africa's fifth-best university, according to the latest rankings by the times higher education related topics more on this story sex for marks scandal: 'my lecturer tried to rape me' top African university probes degree cheats around the BBC Africa today podcasts related internet links Makerere university the BBC is not responsible for the content of external sites.

**Retrieved reference 4**: "Statement: Makerere Visitation Committee lists responsibilities" statement : Makerere visitation committee lists responsibilities the visitation committee appointed by president Yoweri Museveni to look into the affairs of Makerere University has listed its responsibilities, stating that work will be done in a timely and comprehensive manner. A statement by the committee chairperson, Dr Abel Rwendeire, indicates that work started on 17 November with a review of previous reports on the university, and will expand to include consultations with key stakeholders. Makerere University was declared closed indefinitely by president Museveni on 1 November 2016 following a strike by lecturers over unpaid incentive arrears. The university students subsequently joined the strike, demanding to be taught. The committee was thus instituted to look into the issues affecting the university and report after three months. "the committee commenced its work immediately after its inauguration and has already formulated its work plan in five major areas of the university's corporate life namely issues of governance and institutional development; academic affairs; student affairs; staff affairs and financial management," the statement, released on Tuesday, 22 November 2016, reads in part. read the full statement below. Makerere visitation committee statement

**Retrieved reference 5**: "Refugee Law Project — Interactions" refugee law project refugee law project the refugee law project, based at Makerere University in Kampala, works to ensure that all people enjoy their human rights, irrespective of their legal status. It seeks to empower asylum seekers, refugees, deportees, idps and host communities in Uganda to enjoy their human rights and lead dignified lives. The gender & sexuality programme works to ensure that all people access, recover and enjoy their sexual and gender being and rights, and to raise global awareness of the close relationship between violations of sexuality and gender , and patterns of forced migration. It is grounded in the reality that many persons do not access their sexual and gender rights due to social, cultural, political and legal obstacles, while many others lose their rights, as a result of sexual and gender-based violence, particularly in times of conflict, and in the search for asylum. An important emphasis of such engagement has been to highlight the invisibility of men's and boy's experiences of sexual violence, and the extreme Marginalisation and isolation faced by male survivors.

Table 15: Generated results of ReGen (Large) for a specific query with various references. The query and retrieved references are shown in the previous table.

---

**Ground-truth target**: Makerere University, Kampala is Uganda's largest and oldest institution of higher learning, first established as a technical school in 1922. It became an independent national university in 1970. Today, Makerere university is composed of nine colleges and one school offering programmes for about 36,000 undergraduates and 4,000 postgraduate. The main administrative block was gutted by fire in September 2020 and the cause of the fire is yet to be established. U.S. News & World Report has ranked Makerere University as the eighth best university in Africa and the 569th best university worldwide. In the 2020 U.S. News & World Report ranking, Makerere is the highest-ranked university in sub-Saharan Africa outside of South Africa . The Times Higher Education World University Rankings for 2016 ranked it as the fourth best university in Africa. [1] Makerere University was alma mater to many post-independence African leaders, including Ugandan president Milton Obote [2] and Tanzanian presidents Julius Nyerere and Benjamin Mkapa. [3] The former president of the Democratic Republic of the Congo, Joseph Kabila, and former Kenyan president Mwai Kibaki are also Makerere alumni. In the years immediately after Uganda's independence, Makerere University was a focal point for the literary activity that was central to African nationalist culture. Many prominent writers, including Nuruddin Farah, Ali Mazrui, David Rubadiri, Okello Oculi, Ngũgĩ wa Thiong'o, John Ruganda, Paul Theroux, Nobel Prize laureate V. S. Naipaul, and Peter Nazareth, were at Makerere University at one point in their writing and academic careers. Because of student unrest and faculty disenchantment, the university was closed three times between 2006 and 2016. The final time was on 1 November 2016 when president Yoweri Museveni declared it closed indefinitely. [4] The university was reopened in January 2017. [5]
**ReGen without reference**: Makere university school of law is the law school of the makerere university in kampala, the capital of uganda. It is one of the oldest law schools in the country.
**ReGen with 1 reference**: Makere university school of law is the law school of makerere university in uganda. [1]
**ReGen with 2 references**: Makere university school of law is the law school of Makerere university in uganda. It was established in 2002. [1][2]
**ReGen with 3 references**: Makere university, also known as Makerere university, is a public research university in uganda. It is one of the oldest universities in the east african community and the second oldest in the country, after kampala international university. The university was founded in 1922 by the british as a technical training school. It became a full-fledged university in 1962. The annual budget of the university is ug $ 122 million (us $2.2 million), of which the student's contribution accounts for 70% of the total. [1] The university is accredited by the ugandan ministry of education, [2] and is a member of the association of commonwealth universities. Its alumni include several heads of state and government in east africa, including the first president of tanzania, julius nyerere. [3]
**ReGen with 4 references**: Makere university, also known as Makerere university, is a public research university in uganda. It is the oldest and largest university in the country. The name "Makerere" is a compound of the first letters of the names of the ugandan monarchs, mwami and nyerere. The university is located in the city of kampala, the country's capital and largest city. It is accredited by the university of the west african countries, the association to advance collegiate schools of business, and the western association of schools and colleges. It is also one of the few accredited universities in africa to teach the law. [1] [2] The university has produced many notable alumni, including former president of tanzania, julius nyarere, kenya's ngugi wa thiong'o, academics and clergy such as john sentamu, the recently retired anglican archbishop of nairobi. [3] In november 2016, the university was closed indefinitely following a strike by lecturers over unpaid incentive arrears. the university students subsequently joined the strike, demanding to be taught. [4] The visitation committee appointed by president yoweri museveni to look into the affairs of the university has listed its responsibilities, stating that work will be done in a timely and comprehensive manner. [4]
**ReGen with 5 references**: Makere university, also known as makerere university, is a public research university in uganda. It is one of the oldest and largest universities in the country. It was founded in 1922 by king george v of the kingdom of kenya as a technical school for training teachers. The university is located in the city of kampala, the nation's capital and largest city. It is accredited by the commission on higher education, the ugandan ministry of education, and the international association of universities. [2] The university's alumni include heads of state and government in africa, writers ngugi wa thiong'o and john sentamu, [3] former anglican archbishop of nairobi, kenneth nyerere, former president of tanzania juvénal habyarimana and many others. In november 2016, the university was closed indefinitely following a strike by lecturers over unpaid incentive arrears. The university students subsequently joined the strike, demanding to be taught. [4] The visitation committee appointed by president yoweri museveni to look into the affairs of the university has listed its responsibilities, stating that work will be done in a timely and comprehensive manner. [4] Makerere's refugee law project, based at the university, works to ensure all people enjoy the human rights irrespective of their legal status. it seeks to empower asylum seekers, refugees, deportees, idps and host communities in u Uganda to enjoy their human rights and lead dignified lives. [5] An important emphasis of such engagement has been to highlight the invisibility of men's and boys' experiences of sexual violence, and to raise global awareness of the close relationship between violations of sexuality and gender, and patterns of forced migration. Students at the school of law pay an extra 120,000 shillings per semester. [1]

---

