# OpenReview forum: "WebBrain: Learning to Generate Factually Correct Articles for Queries by Grounding on Large Web Corpus"
_ICLR.cc/2023/Conference — Submitted to ICLR 2023_

### Official Review · Reviewer_Ej3Q · 2022-10-23

**Confidence:** 4
**Correctness:** 3
**Technical Novelty And Significance:** 2
**Empirical Novelty And Significance:** 2
**Recommendation:** 5

**Clarity, Quality, Novelty And Reproducibility:**

This paper was reasonably clear, and I thought that the setup was pretty solid. It's an entry in a long line of work on generating Wikipedia paragraphs from references or other external knowledge.

**Strength And Weaknesses:**

Strengths:
- This is an interesting and important task, and I think it's one that could be a fruitful combination of lot of existing work on both generation and understanding.

Weaknesses:
- Despite the grandiose motivations, the authors consider an extremely narrow instantiation of this task:
  - Only the first paragraph is used as the generation target, rather than entire articles. When I read the abstract and introduction I was excited to see how the authors would handle generation / understanding of long sequences, since this is an area where I think this benchmark would be particularly useful, but was disappointed to see that they largely punted on this problem.
  - Furthermore, the authors don't use the entirety of the reference articles, instead taking only a single paragraph per article that is specifically chosen to maximize informativeness, which I think gives a bit of an unfair advantage to the retrieval-based systems (since they're much more likely to retrieve something useful).

- In general, I feel like the paragraph "Reference Passage Selection " in 3.2 completely misses the mark. In particular:

> The original Wiki articles and reference articles tend to be very long. To adapt the capacity of most pre-trained language models (e.g., BERT has a 512 token limit), we use the first section of Wiki articles as the generation target, and we select the passage from the reference article with the highest PST value as the supporting passage.

I'm not sure that I feel very good about our benchmarks having to be "adapted" to be consumable for existing models while trading off the essence of the task (synthesizing information from multiple long documents to generate a long document). We should set the benchmarks as goals, see how current models and techniques perform on them, and allow them to be targets for future work.

- I feel like the baselines are pretty weak, and the ReGen model is a pretty standard retrieve-and-generate baseline. I don't think there's too much modeling novelty here.

- Furthermore, the query distribution seems pretty unnatural. For example, in the appendix, the query used is "how google translate work". This is a query with a pretty clear information-seeking / question intent that can be answered by a paragraph (or paragraphs). However, the wikipedia page for "Google Translate" answers far more than just this question, it contains a variety of general information about the query "google translate" (which itself could have many different question intents). So, I feel like the source of supervision provides a different sort of information than what a query like this would actually want (let alone that in practice, only the first passage is used, which contains even less information because of the way that lead sections in wikipedia are structured to contain general high-level information: https://en.wikipedia.org/wiki/Wikipedia:Manual_of_Style/Lead_section).
  - The original motivation mentioned "unseen factual queries", but it's not really clear that generating an article is what you want for this. If it's factual, you can probably answer it with a paragraph or two. I think this work would benefit from a clearer exposition of what these queries would actually be, and what the right source of supervision / data for effectively addressing these queries would look like.

**Summary Of The Paper:**

This paper introduces a new benchmark for generating the first paragraph of wikipedia articles (with references) from supporting paragraphs. The authors compare a retrieve-and-generate baseline against several non-retrieval baselines (e.g., BART, GPT-2, and FiD + BART). None of these methods (including GPT-3) perform well on the task, suggesting room for future improvement.

**Summary Of The Review:**

While I like where this paper is going, I feel like the practical artifact and problem tackled here are a bit too small-scale and narrow. We're at a point in NLP where things actually kind of (?) begin to work, and there have been a variety of impressive results on generating and understanding long documents. I think this work would really shine in pushing this direction, but instead, it compromises itself to fit the limitations of existing models, limiting its usefulness for future work. Thus, I'm not sure that I can recommend acceptance.

---

> ### Author Response · Authors · 2022-11-12
> **Response to Reviewer Ej3Q (Part 1 of 2)**
>
> Thanks for taking the time to thoroughly read our paper and providing the constructive comments. We are glad that you find our proposed task interesting and important. Below, we address your comments on weakness point-by-point and explain task setting, which seems to be your primary concern.
>
> **Comment 1:** Generating the first section of Wikipedia is an extremely narrow instantiation.
>
> **Response:** We fully understand your concern. Generating the complete Wiki page is the ultimate goal of this line of research. Even though we set WebBrain to generate the first Wiki section, the line of research is not limited to it. However, we also understand that generating a complete Wikipedia page is an ambitious goal that involves more aspects. For example, we need to maintain consistency among the contents under different sections and  avoid redundancy. We have done some preliminary experiments in generating Wikipedia pages by using models to generate sub-topic queries given the input query, and recursively generate each section. Even though some of the results seem promising, it is clear that the task is very challenging and beyond the reach of the current approaches. We have added a description about generating a whole page with multiple sections in the revised paper (see Appendix H).
>
> Due to the huge difficulties of the full task, we designed a simpler task to generate the first section. This simpler task is not set as the ultimate task but as a good start in this line of research: generating a text to describe the retrievable information about a topic. Our motivation behind the setting is to see how well this limited task can be handled by the current techniques and what remaining problems we may face before targeting a more ambitious task. As we can see in the experiments, generating the first Wiki section is already a very challenging task for the current models. It would be too ambitious to target the generation of full pages right away. Targeting a too difficult problem would also generate much less interest from the researchers to work on the problem. So, our work is a good compromise to attract research efforts toward the ultimate goal of generating more complex pages. For a more detailed explanation, please refer to our general response 2.
>
> **Comment 2:** Reference passage selection gives an unfair advantage.
>
> **Response:** The main reason we perform reference passage selection is for experiment efficiency. In other words, we need to run a single experiment iteration within a reasonable time; otherwise, the task would be infeasible for us and for most researchers. Instead of randomly sampling a subset, we include the passages that are more likely to provide useful evidence for the topics. This is done to reduce the chance that we try to create an article based on a set of irrelevant documents - a situation that may occur in practice, but is not the main focus of this work. Instead, we want to propose the task of generating a section with a reasonably relevant set of texts.
>
> To address your concern, we further conducted an experiment by using reference passages retrieved from a **full passage set** to generate the target. The full passage set contains about 204M passages, which is challenging for retriever. As a comparison, we also show the performance with reference passages retrieved from the **selected passage set** (containing only passages selected by our strategy). The results are shown below (for full results, please refer to Table 4 and Table 8 in our revised manuscript).
>
> | Model            |   BLEU-1  |   METEOR  |  ROUGE-L  |   CIDEr   | TripleScore |
> |------------------|:---------:|:---------:|:---------:|:---------:|:-----------:|
> | ReGen (Selected) |   24.97   |   13.38   |   20.97   |   19.87   |    10.99    |
> | ReGen (Full)     | **27.81** | **14.89** | **22.00** | **19.40** |  **13.68**  |
>
> We can observe that ReGen with passages retrieved from the full set performs better. This reflects that ReGen's retriever can indeed find useful evidence for generation. Besides, this also indicates that **the reference passage selection would not bring an unfair advantage** to retrieval-based systems. In contrast, the full passage set can provide more topic-relevant passages, which can further facilitate the generation.
>
> Please also refer to general response 3 for more explanation.
>
> **Comment 3:** Baselines are weak, and ReGen lacks novelty.
>
> **Response:** The baselines we used are typical methods for retrieval-augmented text generation, which have been used in various studies. Besides, we do not claim that ReGen's architecture is the main contribution to the paper, compared to the task definition and dataset of WebBrain. The model is designed and tested to show how the current techniques can, or cannot, reasonably deal with this problem.
>
> Please refer to general response 1 for more details.

---

> > ### Author Response · Authors · 2022-11-12
> > **Response to Reviewer Ej3Q (Part 2 of 2)**
> >
> > **Comment 4:** The definition of what the factual query would be is unclear.
> >
> > **Response:** Sorry for the confusion caused by the query in the demo screenshot. We would make it clear that the factual queries in the WebBrain task are entity queries (e.g., titles for Wikipedia). The "unseen factual queries" refer to newly emerging entities (e.g., the name of a newly founded academic institute) that are not included in Wikipedia. In both the WebBrain dataset and model experiment, we use a Wikipedia title as the input query. The example query "how Google Translate work" in the demo screenshot is to show how ReGen would be used for more natural questions. In the revised paper, we also added the demo result of "Google Translate" for comparison (see Appendix H).
> >
> > We also agree that various intents may be involved in a query. We assume that the information intended by the user is the type of general information summarized in the lead section of a Wikipedia page. According to (https://en.wikipedia.org/wiki/Wikipedia:Manual_of_Style/Lead_section), "The lead is the first thing most people will read upon arriving at an article, and may be the only portion of the article that they read". This shows the practical value of generating such a lead section for a new topic.

---

> > > ### Comment · Reviewer_Ej3Q · 2022-11-28
> > > **Response to authors**
> > >
> > > Thanks for the thorough response and for answering some of my questions, especially providing the results for using reference passages from the full set.
> > >
> > > I think that this work is overall quite solid, but for a paper whose primary contribution is a new benchmark and task, I feel like it aims a bit too low---I think this paper would be far more interesting if it addressed the problem of generating and evaluating longer documents, whereas the current instantiation doesn't feel like it pushes the state of the field too much beyond providing another generation dataset. I'll raise my score to marginally below reject threshold, since the paper is well-executed, but I think could be reasonably asked to be more ambitious.

---

> > > > ### Author Response · Authors · 2022-11-30
> > > > **Thanks for your comment**
> > > >
> > > > We are glad that our response has helped to alleviate some of your concerns. We also appreciate your score improvement. Actually, our dataset can support the full Wiki document generation, though in this paper we focus on the short article generation, which is an essential subtask towards this goal. We agree that generating a full article is much valuable yet challenging. We will continue to explore it regarding both method and evaluation. Thank you.
> > > >
> > > > Best,
> > > >
> > > > Authors

---

### Official Review · Reviewer_892z · 2022-10-25

**Confidence:** 4
**Clarity, Quality, Novelty And Reproducibility:** 1. Some parts of the paper are not ve…
**Correctness:** 3
**Technical Novelty And Significance:** 3
**Empirical Novelty And Significance:** 3
**Recommendation:** 6

**Strength And Weaknesses:**

Strength
1. The paper introduces WebBrain, which lets the model retrieve supporting evidence and generate factual articles given a factual query. The proposed dataset is somewhat similar to the Wizard of Wikipedia (Dinan et al., 2018). The newly proposed dataset is interesting and large-scale. The authors crawled and cleaned Wikipedia. The proposed task and corresponding dataset are very interesting and worthy of future research.

2. The paper proposes a new retrieval-augmented generation framework based on SPLADE and FiD. The proposed methods achieve the best results over automatic and human evaluation. The experiment section is very comprehensive. The authors conduct an ablation study with different retrieval models and show the impact of the different numbers of retrieved references. The paper also checks the impact of a number of references. Those results are clearly represented in tables or charts with detailed explanations. The paper shows the case study, human evaluation, and reference mark correction strategy in the appendix.

Weaknesses
1. The paper uses n-gram overlapping metrics for automatic evaluation. The paper needs to include some newer metrics such as BERTscore (Zhang et al., 2019), and BARTScore (Yuan et al., 2021) which can check semantic similarity.

2. Most of the experiment analyses are in quantitative way. I would like to see more qualitative analysis.



Zhang, T., Kishore, V., Wu, F., Weinberger, K. Q., & Artzi, Y. (2019). Bertscore: Evaluating text generation with bert. arXiv preprint arXiv:1904.09675.

Yuan, W., Neubig, G., & Liu, P. (2021). Bartscore: Evaluating generated text as text generation. Advances in Neural Information Processing Systems, 34, 27263-27277.



Dinan, E., Roller, S., Shuster, K., Fan, A., Auli, M., & Weston, J. (2018). Wizard of Wikipedia: Knowledge-powered conversational agents. arXiv preprint arXiv:1811.01241.

**Summary Of The Paper:**

This paper introduces a new task called Web-Brain which aims to generate short factual articles for queries by mining supporting evidence from Web. The paper also proposes a new large scale dataset with English Wikipedia. The paper also provides a new framework called ReGen based on SPLADE and FiD. The model is evaluated with both n-gram overlapping metrics and factual correctness metrics. The paper analyze the impact of retrieval, number of references in a quantitative way. The paper also did both human and automatic evaluation.

**Summary Of The Review:**

Overall, the paper proposes a new interesting task with a corresponding large-scale Wikipedia-based dataset. The experiment part is quite comprehensive.

---

> ### Author Response · Authors · 2022-11-12
> **Response to Reviewer 892z**
>
> We appreciate the reviewer's useful suggestions for evaluating the model. We are pleased that you confirmed the value of our proposed task and the corresponding dataset. We have addressed each of your comments on weakness and revised our manuscript based on your suggestions.
>
> **Comment 1:** It should include new metrics for automatic evaluation.
>
> **Response:** Following your suggestion, we have added the results of BERTScore and BARTScore in Table 11 of our revised manuscript. The results are shown as follows.
>
> |                      |          | BERTScore |          |     BARTScore    |
> |:---------------------|:--------:|:---------:|:--------:|:----------------:|
> | Model                |     P    |     R     |    F1    | (log-likelihood) |
> | BART (Base)          |   96.8   |    94.4   |   95.5   |      -4.375      |
> | BART (Large)         |   97.6   |    94.8   |   96.2   |      -4.280      |
> | GPT-2                |   87.3   |    85.4   |   86.3   |      -4.560      |
> | FiD+BART (Base)      |   96.2   |    93.9   |   95.1   |      -4.091      |
> | FiD+BART (Large)     |   97.6   |    94.9   |   96.2   |      -4.014      |
> | ReGen (Base)         |   96.2   |    93.9   |   95.0   |      -4.071      |
> | ReGen (Large)        | **97.7** |  **97.8** | **97.8** |    **-3.936**    |
> |                      |          |           |          |                  |
> | ReGen (w/ Retriever) |   90.2   |    84.5   |   87.3   |    **-4.303**    |
> | ReGen+BM25           | **91.6** |    91.3   | **91.2** |      -4.326      |
> | ReGen+DPR            |   83.3   |  **91.4** |   87.1   |      -4.664      |
> | ReGen+SPLADE         |   88.1   |    85.5   |   86.8   |      -4.381      |
>
> We can see the results are generally consistent with the ones in Table 3 (please refer to our paper). Our ReGen (large) can achieve the best result in terms of semantic similarity with the ground-truth target. We also provide the results of ReGen with various retrievers. More discussions on these results have been presented in Appendix E.
>
> **Comment 2:** It is better to perform more qualitative analysis.
>
> **Response:** Thanks! We have added more qualitative analysis to the case study (Appendix G) per your advice. In this case study, the generated results of ReGen are compared to those of several baseline methods. This comparison is performed under two scenarios where the reference is either given or not. Furthermore, we also show some generated results of ReGen with varying numbers of retrieved references. We believe these analyses will provide readers with a more intuitive understanding of the generated results, the remaining problems, and the impact of the retrieved references.
>
> **Comment 3:** How to construct WebBrain-R and WebBrain-G is unclear.
>
> **Response:** Thanks for your advice. Due to the limited space, we have added more details regarding the dataset construction of WebBrain-R and WebBrain-G in Appendix A.
>
> **Comment 4:** It should provide code for reproduction.
>
> **Response:** Thanks for your comment. We cleaned up the code and provided it with an anonymous link for review (https://anonymous.4open.science/r/WebBrain/). We have revised the paper and added this link to the implementation details in Appendix A.

---

> > ### Comment · Reviewer_892z · 2022-11-17
> > **Thank you for your response**
> >
> > Thank you for your comments. I think the authors addressed all of my concerns.

---

### Official Review · Reviewer_dCpa · 2022-10-26

**Confidence:** 4
**Correctness:** 3
**Technical Novelty And Significance:** 2
**Empirical Novelty And Significance:** 3
**Recommendation:** 8

**Clarity, Quality, Novelty And Reproducibility:**

The paper is clear and good quality. The system is described in detail and should be reproducible.

The novelty of the approach is mainly in the size of the released corpus and in including retrieval as part of the task, together with multi-document summarization.

The techniques proposed by the authors are very interesting, but they are only applied to the dataset they created, so it's not totally clear how their modeling choices would stack up against other methods on a more competitive benchmark.

**Strength And Weaknesses:**

Strengths:
* The task examined in this paper (retrieval + multi-document summarization) is important.
* The dataset released by this paper is much larger than comparable datasets and could be useful for furthering work on this topic.
* The authors describe in detail many technical details of ReGen. These details can be useful to practitioners for reproducing ReGen’s results and in their own work.

Weaknesses:
* Limiting the generation to only the first passage of wikipedia pages is a pretty strong limitation, also making this work very close to existing multi-document summarization works. The authors do acknowledge this similarity though.
* In “Reference Passage Selection” and “Dataset Generation”, the authors select input passages for training with simple word overlap or BM25. It seems like this could be easily improved by using an entailment model, a dense retriever or even SPLADE.
* The proposed "WebBrain-Raw" dataset would be more interesting if it was multilingual instead of English-only.

**Summary Of The Paper:**

The paper proposes a new task called “WebBrain”. The objective of the task is to learn to generate a fluent, informative and factually correct short article from a query given some search results. The task is essentially a combination of 2 components:
  1. retrieval of evidence passages given a wikipedia page title,
  2. multi document summarization of the retrieved evidence with the first paragraph of the wikipedia page as target.

The authors contribute a dataset "WebBrain-Raw", comprised of English wikipedia plus all crawlable references. The dataset is cleaned with an interesting set of heuristics.

The authors additionally propose a baseline system called "ReGen". ReGen combines several modeling choices:
  * training a SPLADE retriever for this task with hard negatives are mined,
  * tuning sparsity in document representations for retrieval,
  * filtering retrieved documents trading off consistency and diversity,
  * fusion-in-decoder for generation,
  * making generation more grounded in the references, by pre-training the generation component to generate individual sentences from a reference passage at time.

The authors finally present analysis showing the usefulness of these modeling choices, compared to more vanilla choices.

**Summary Of The Review:**

This looks like a useful dataset+baseline contribution for the important task of multi-document summarization. The techniques used by the authors seem solid.

---

> ### Author Response · Authors · 2022-11-12
> **Response to Reviewer dCpa**
>
> Thanks for your time and effort on our paper. We are glad that you find our examined task is important for the area and you think that the released dataset and model would be valuable for future work on this topic. Below, we respond to your comments on weakness point-by-point.
>
> **Comment 1:** Generating the first section of a Wikipedia page is a limitation.
>
> **Response:** We agree with you that it is a limitation to generate only the first section. The task is designed this way due to the following reasons:
>
> 1. Generating the first section is **a reasonable start** to exploring generating the entire Wikipedia pages, as the content of the first section is comprehensive and its length mostly suits the token limit of current pre-trained models. Through the experiments in the paper, we can also see that generating the first section is **still challenging** for current techniques.
> 2. Instead of a Retrieval-Generation framework, we think that generating the complete Wikipedia pages involves more modules and systematic designs such as section title generation, table generation, etc. We also did some preliminary experiments on generating full Wikipedia pages with different sections. In the experiments, we use another model to generate the section titles and recursively generate different sections conditioned on the section titles. This preliminary study tried to gauge the feasibility of such a task. It turned out that the task is too challenging for now, and the interest from researchers to work on the task now is likely to be small. Once we can handle the limited task well, we could extend it to the task of generating full pages in the future. In fact, the WebBrain-Raw dataset contains the complete Wiki pages and the reference articles, which support using the complete Wiki pages as the generation target.
>
> **Comment 2:** Applying entailment models for reference passage selection may bring further improvement.
>
> **Response:** Thanks for your helpful comment. In the reference passage selection, we apply a simple heuristic based on word-overlapping, which is commonly used in many NLP tasks (such as open-domain QA). We agree that applying domain-adapted entailment models for reference passage selection would increase selection accuracy. We would consider this in our future exploration.
>
> **Comment 3:** Involving more languages in "WebBrain-Raw" is interesting.
>
> **Response:** Thanks for your advice. This would be a very interesting direction for future research. Currently, although our dataset is built on English Wikipedia and our model is based on the English articles, we have built a preliminary end-to-end application/demo that integrates a translation model. The original query in other languages can be translated into English, and the generated short article is then translated back to the query language. We find that the generation quality is reasonably good. We agree that training on multilingual data would be more interesting, and we will continue working on this in our future work.

---

> > ### Comment · Reviewer_dCpa · 2022-11-30
> > **Thank you for your response**
> >
> > I agree with some of the points raised by other reviewers that the work described in this paper is well executed but could reasonably be asked to be more ambitious.

---

### Official Review · Reviewer_TtTt · 2022-10-28

**Confidence:** 4
**Correctness:** 3
**Technical Novelty And Significance:** 3
**Empirical Novelty And Significance:** 3
**Recommendation:** 6

**Clarity, Quality, Novelty And Reproducibility:**

#### Clarity & Reproducability
The paper is clearly written and all parts are pretty well described. The authors commit to releasing the data, which will allow easy replication.

#### Novelty
The dataset is similar in form to previous work, but extends that previous work to the scenario where references must be retrieved from a very large corpus. Similarly, the model is a small modification of existing approaches but it appears to work better than well chosen baselines for this task.

#### Quality
There are a lot of details in the filtering procedures used to select the contents of the dataset, and assign references to sentences. These are all justified int the text, but are not supported by any sort of analysis. If this is going to become a benchmark dataset, going forward, it would be good to see a discussion of how these choices affect the task. In particular, see my question about test target leakage into the retrieval corpus above (under Weaknesses).


**Details Of Ethics Concerns:**

This work promises the release of 260M non-Wikipedia documents without any discussion of potential copyright or privacy issues. Both of these should be addressed. The authors could also consider methods of supporting user data removal requests or, alternatively, release pointers into a store like CommonCrawl, which has support for data removal requests.


**Strength And Weaknesses:**

#### Strengths
- This paper promises a massive new dataset for retrieval augmented text generation, that could be very useful to the NLP community. Having said that, I do have questions about the release with respect to copyright and privacy concerns (detailed below in the ethics section).
- The end system can retrieve references, and link them to generated sentences, which is a very nice end product which could have real utility beyond unconstrained text generation, which cannot provide any form of attribution for its predictions.
- There are a number of interesting modeling and training choices made, which lead to significant improvements over a very large language model (GPT3) and also similar retrieval agumented approaches with different choices of retriever and training procedure.

#### Weaknesses
- There are a lot of separate contributions here that are not independently evaluated (data filtering / retrieval filtering / warmup strategy). It would be nice to see evaluations that validate these choices.
- The reference passage filtering process selects the references included in the retrieval corpus references according to term overlap with the target Wikipedia article. If I'm understanding this process correctly, this means that the references stored in the corpus have been selected according to observations of the test time targets, so there is some leakage of information from the test targets into the model input. The paper should discuss this.
- The prompt given to GPT3 "Introduce [Page Title]" does not mention that the target is Wikipedia-style text, and the example in Figure 1. does not look much like the start of a Wikipedia article. Meanwhile, GPT3 is definitely able to generate Wikipedia style text if prompted to do so. The comparision to GPT3 would be stronger if the prompt was better tied to the actual task.


**Summary Of The Paper:**

This paper presents a new dataset and associated task that involves generating the first section of Wikipedia pages from a set of retrieved references. The dataset is similar in form to previous work like WikiSumm, but it is significantly larger and the authors promise to release the references, which have been downloaded from non-Wikipedia sites.

The dataset generation process involves a number of filtering steps, based on term overlap, that narrow down the reference data to passages that are likely to be related to the target Wikipedia article and associate these passages with sentences in the target. The goal, at test time, is to retrieve reference passages that may be related to a topic, and then
generate the target section of the Wikipedia article.

Along with the dataset, this paper presents a new model, ReGen. ReGen is composed of a retriever (based on SPLADE) and a reference encoder and target decoder (based on FiD). When trained on the WebBrain data, ReGen outperforms both large language model baselines and also a retrieval augmented model with a dense retriever (FiD + BART) according to a range of automatic metrics, and also annotator judgements of a small set of predictions. I have some questions about some of these comparisions below, but it is clear that ReGen is capable of generating coherent and informative text. It is also capable of providing references for this text, which is a significant advantage over methods that don't include retrieval.

There are a number of nice ablations that show how the different systems' performances change with different numbers of retrievals and gold references. These comparisons are useful in understanding the tradeoffs between retrieving a lot of supporting evidence, with lower precision, vs fewer high quality references.

**Summary Of The Review:**

This paper presents a significant new dataset and task that could be useful to the community, going forward, in benchmarking methods of retrieval augmented generation. The new model is also quite different from previous work, which has generally relied on dense DPR-style retrievers.

Overall, I think this paper is a nice contribution and I am currently leaning toward acceptance. However, I also have some serious questions below about the data release strategy, and how the authors propose to handle concerns about releasing multiple terrabytes of data which may include copyrighted works or personal information.

---

> ### Author Response · Authors · 2022-11-12
> **Response to Reviewer TtTt (Part 1 of 2)**
>
> We appreciate the reviewer's insightful comments and the generally positive evaluation of the paper. We believe that addressing these points in the paper would indeed improve its quality. Below are our responses to the comments.
>
> **Comment 1:** It would be nice to evaluate each contribution separately.
>
> **Response:** Thanks for your suggestion. The data/retrieval filtering process is empirically performed according to our preliminary study on a smaller-scale subset. For example, we remove sentences like "You can help Wikipedia by expanding it" because we find they frequently appear in the generated text. Removing it can directly improve the informativeness in some cases. We have not evaluated these processing strategies on the full corpus because of the large volume of data: our model training takes a long time, and it was difficult to compare and evaluate different options. As for the warm-up strategy, we took it as a common approach to increasing the performance of ReGen. Its effectiveness can be inferred from the performance improvement of ReGen over FiD+BART. It would indeed be interesting to investigate other alternative approaches for the task based on the dataset.
>
> **Comment 2:** There may be information leakage in reference selection.
>
> **Response:** We understand your concern, but we do not think the procedures we take would lead to information leakage. For **WebBrain-R**, our reference passage selection can largely reduce the training data (from 204 million to 3.2 million passages) and ensure that the corpus can provide supporting evidence for generation. After the filtering, we keep a subset to a reasonable size (million-level), which can accelerate model training and allow more researchers to investigate the task. As for **WebBrain-G**, our passage selection can ensure that the provided references are relevant (can be treated as *oracle* references), thus helping the model learn to extract useful information from the references. Such filtering may not be possible in practice, but it is a common approach used to build a workable dataset in many previous studies.
>
> To further address your concern, we have conducted a full pipeline test, which is closer to real-world scenarios. In this test, the retriever needs to find evidence from a **full passage set** (built on all passages in references, about 204 million passages in total). Then, the generator has to generate target text based on these **retrieved passages**. The results are shown below (the full results are in Table 4 and Table 8 of our revised manuscript).
>
> | Model            |   BLEU-1  |   METEOR  |  ROUGE-L  |   CIDEr   | TripleScore |
> |------------------|:---------:|:---------:|:---------:|:---------:|:-----------:|
> | ReGen (Selected) |   24.97   |   13.38   |   20.97   |   19.87   |    10.99    |
> | ReGen (Full)     | **27.81** | **14.89** | **22.00** | **19.40** |  **13.68**  |
>
> From the results, we can observe that our ReGen can generate better results with references retrieved from the full passage set. This indicates that the full passage set can provide more relevant passages. Besides, the results also show that ReGen can indeed make use of the retrieved references (rather than the *oracle* references) for generation.
>
> We will also release the full passage set. Researchers can perform a more in-depth study on this set if sufficient computing resources are available. We have added more discussion about the motivation to apply passage selection in Appendix D. Please also refer to general response 3 for more explanation.
>
> **Comment 3:** Other prompts for GPT-3 can perform better.
>
> **Response:** Thanks for your suggestion! For GPT-3, we tested some other prompts that directly specify to generate Wikipedia-style text. The best result is obtained by using the prompt "Introduce [X] in Wikipedia-style". As you noted in your comment, this prompt can lead to better performance in both naturalness and factualness (see the first row and the second row). Moreover, ReGen can still achieve the best performance (see the third row). We have revised the description and the figure in the section "Performance vs. GPT-3" based on the new result. More discussions about using different prompts in GPT-3 have been added to Appendix F.
>
> | Model                   | BLEU-1 | BLEU-4 | METEOR | ROUGE-L | CIDEr | QAGS  | TripleScore |
> |:------------------------|:------:|:------:|:------:|:-------:|:-----:|:-----:|:-----------:|
> | GPT-3 (Original Prompt) | 17.24  | 2.19   | 9.14   | 13.57   | 6.70  | 17.00 | 0.29        |
> | GPT-3 (New Prompt)      | 19.72  | 3.60   | 9.93   | 14.89   | 9.33  | 18.16 | 2.08        |
> | ReGen (Large)           | **28.94**  | **8.74**  | **14.87**  | **21.27**  | **19.30** | **48.72** | **12.56**        |

---

> > ### Author Response · Authors · 2022-11-12
> > **Response to Reviewer TtTt (Part 2 of 2)**
> >
> > **Comment 4:** There are some ethics concerns about the dataset.
> >
> > **Response:** Thanks for your reminder. We are considering how to release the dataset. We will require the user to sign an application form in which we list the terms of use (ToU) to limit the application scope and address privacy and copyright issues. Currently, we have two  plans for data release. The first one is to release the 260M non-Wikipedia data in URL-format. Along with the dataset, we will provide our crawl tools (similar to WikiSUM). We think only providing URLs can well-address the privacy and copyright issues, but it makes it harder for researchers to get the dataset as crawling such a volume of data consumes a lot of resources, and some webpages might be unreachable after a while. Another way is to release the complete dataset, which consists of 154 GB-sized file chunks (and is easier to download). To mitigate privacy- and copyright-related risk, we can follow the ToU of CommonCrawl (https://commoncrawl.org/terms-of-use/) and remove the specific data from our raw dataset when we receive a data removal request. As you suggested, constructing pointers into a public web store could also be a good solution. We are trying this, but it will take some time due to the huge volume of web stores like CommonCrawl.

---

### Author Response · Authors · 2022-11-12
**General response to all reviewers (Part 1 of 2)**

We thank all reviewers for their time and effort on our paper. Below, we would like to address your common concerns.

**General response 1:** declare our motivation and contribution

In this work, we aim at generating factually correct articles for queries supported by some references. As an initial work, we focus on generating a short article (like the first section of a Wikipedia page) by retrieving supporting evidence from a large Web corpus. This task can be used to explore two key components, namely the retriever and the generator. The proposed task is a start toward achieving the ultimate goal of generating a full Wikipedia page on a topic. On the other hand, our study is an attempt of extending existing information retrieval methods to information aggregation methods (from search to generation) to try to address this problem. We believe the generation task has its huge practical value in the current era of information explosion.

The main contribution of this paper is the task definition of WebBrain and the construction of the corresponding datasets. We plan to release WebBrain-R with a full passage set, WebBrain-G, and WebBrain-Raw. They can support not only our WebBrain task but also further research. From a model perspective, our contribution lies in choosing the proper technique combination from current models for the new task (e.g., using SPLADE rather than DPR), and highlighting promising directions in ReGen that can effectively improve the model's performance (e.g., model warm-up and reference mark correction). We do not claim that our model is the one that fits the task. Rather, it is just a possible way to deal with the task, and there is a need to develop more adapted models for it.

**General response 2:** explain the reason of choosing the first section as the generation target

In our original submission, we briefly discuss this issue in the paper limitation. We would further address the reviewers' concerns by answering the following questions.

**Q1:** Why do we choose to generate the first section of Wikipedia?

**A1:** We think generating the complete Wikipedia page, while very attractive, would be a too ambitious goal to address from the beggining. This would involve a broad range of NLP techniques and system designs. For example, we need to consider how to select proper section titles (sub-topics); how to retrieve proper references given the input query, the section title, and the generated context; and how to maintain the consistency/diversity of different sections, etc.

We consider generating the first section a reasonable task to address at the beginning of the long research pathway. The first section of Wikipedia contains comprehensive descriptions for the entity query, which can help the user quickly understand general knowledge. The fact that many users would limit themselves to reading this section clearly shows the practical value of the task.
Therefore, generating the first section is a proper choice for us to explore generating short factual articles. Besides, according to our experiments, generating the first section of Wikipedia is very challenging. The generated text is far from being perfect. It would be interesting for researchers to work on the problems observed in the limited task before moving to a broader and more ambitious task.

**Q2:** Have we tried to generate the complete Wikipedia page?

**A2:** Yes, we conducted some initial experiments on generating a complete Wikipedia page when we designed the task. Concretely, we first utilized a sub-topic generation model to generate the section titles for the input query. We then fed the query and the section titles into our ReGen recursively, and obtained a long Wiki article comprising several sections. Such a method worked well for some queries but generated redundant texts under different sections as we did not use the section title during the training. According to the experimental results, we judged that the generation of a full page is too ambitious for now. This is why we designed the simpler task of genetating the first section. We did not include this in the original submission because of the length limit and because we thought it could be only loosely related to the main topic of the paper. Given the comments of the reviewers, we have added this exploration and a generated page sample in Appendix H (System Demonstration).

---

> ### Author Response · Authors · 2022-11-12
> **General response to all reviewers (Part 2 of 2)**
>
> **General response 3:** explain the use of reference passage selection strategy
>
> We use a simple heuristic to select the most relevant passage from the reference. We would like to explain this process by answering the following questions.
>
> **Q1:** Why not include all the reference passages into the WebBrain-R's corpus?
>
> **A1:** The WebBrain-R is a working dataset for training. Considering efficiency, we need to conduct experiments on WebBrain-R that take a reasonable amount of time. As our retriever works at the passage-level, we need to split the original long documents into passages. According to our statistics, after data cleaning and filtering, there are at least 204 million passages in total, which is a great challenge for the index construction and hardware resources (requiring around 800 Tesla V100 32G GPU hours and 1,200 GiB of memory). Therefore, we keep a subset with reasonable size (million-level) to train the model, which is a size more manageable by other researchers to investigate the task.
>
> **Q2:** Would it lead to information leakage or an unfair advantage to the retrieval model?
>
> **A2:** No. To address this concern, we have conducted further experiments on generating texts based on passages retrieved from a **full passage set**. There is no additional selection process for these retrieved passages. The results are listed below.
>
> | Model            |   BLEU-1  |   METEOR  |  ROUGE-L  |   CIDEr   | TripleScore |
> |------------------|:---------:|:---------:|:---------:|:---------:|:-----------:|
> | ReGen (Selected) |   24.97   |   13.38   |   20.97   |   19.87   |    10.99    |
> | ReGen (Full)     | **27.81** | **14.89** | **22.00** | **19.40** |  **13.68**  |
>
> We can observe:
>
> - Performance of ReGen (Full) is better, indicating that the full passage set can provide more relevant evidences, and ReGen's retriever trained on WebBrain-R is able to find them. This verifies that our reference passage selection is a simple yet effective strategy for retriever training. Furthermore, according to our experiments in Appendix D, we find that the performance of all the retrievers has a similar tendency on both the full passage set and the selected passage set. Training and evaluating models on WebBrain-R can well reflect their performance on the full passage set.
> - ReGen's generator can perform well with the **retrieved** passages, though it is trained with the **oracle** reference passages. Therefore, learning on WebBrain-G can provide good generalizability for models. Besides, the better performance of ReGen (Full) over GPT-3 (see Table 11 in our paper) also verifies the superiority of retrieving explicit knowledge for generating informative and factual texts.
>
> Finally, we plan to also release the full passage set to support in-depth research. It requires more computing resources but has the potential to find more relevant passages.
>
> According to the reviews, we have **revised our paper** as follows:
>
> - We add the construction process of WebBrain-R and WebBrain-G in Appendix A.
>
> - We add an experiment to test ReGen's performance with references retrieved from the full passage set. More discussion is presented in the section "Impact of Retrievers" and Appendix D.
>
> - We add BERTScore and BARTScore as evaluation metrics. The results and discussions are presented in Appendix E.
>
> - We revise the performance comparison with GPT-3 by using a more Wiki-relevant prompt. The results and analysis are presented in Appendix F.
>
> - We conduct a more comprehensive qualitative analysis in Appendix G. The impact of references is discussed.
>
> - We revise the system demonstration section in Appendix H to show three scenarios: using an entity as the query; using a specific question as the query; and generating a full Wikipedia page.
>
> We believe these revisions, based on the comments, have indeed improved the quality of our paper. Thanks again for your time in reviewing our paper, and we hope our responses have addressed your concerns.

---

### Author Response · Authors · 2022-11-24
**Looking forward to your feedback**

Dear Reviewers,

Thanks for your valuable and constructive comments, which greatly help us improve our paper.

We have provided detailed responses and revised the paper accordingly. We would appreciate it if you could let us know if our responses have addressed your concerns and whether you still have any other questions on the current draft and our rebuttal.

Best,

Authors

---

### Decision · Program_Chairs · 2023-01-20

**Decision:**

Reject

**Justification For Why Not Higher Score:**

(see weaknesses above)

**Justification For Why Not Lower Score:**

n/a

**Metareview: Summary, Strengths And Weaknesses:**

This paper presents a large scale dataset of Wikipedia articles and the references contained therein. The task is to generate the first section/paragraph of the article given the query (the title of the Wikipedia page). The authors claim that their work generates factually correct information. The method contains two steps: (1) First retrieve a set of documents that are relevant for the given query, (2) use information in these documents to generate factually correct passages.

Strengths
* well-written paper
* release of dataset
* experiments with a bunch of different eval metrics

Weaknesses
* no human evaluation of factuality
* no modeling innovation to enforce factuality
* no interesting processing of data apart from simple crawling and cleaning
* unclear what are the main differences from retrieval augmented approaches
* generating only the first section is pretty limiting as an application
* unclear why certain decisions are made throughout the paper without further details on these (heuristics for reference selection, data cleaning etc.)
* a lot of queries are simply noun compounds that don't display any special linguistic behavior
* why is BLEU-4 score missing from certain tables?

**Summary Of Ac-Reviewer Meeting:**

The reviewers agreed that the paper in its current form is not very useful to the community both in terms of the utility of the released data and the application of generating the first section of Wikipedia. The reviewers agreed that execution was good, but the paper lacks any strong data or modeling contribution that has a takeaway message for the community.